# Microbiome differential abundance methods produce different results across 38 datasets

Jacob T. Nearing [1,7 ✉], Gavin M. Douglas[1,7], Molly G. Hayes [2], Jocelyn MacDonald[3], Dhwani K. Desai[4], Nicole Allward[5], Casey M. A. Jones[6], Robyn J. Wright[6], Akhilesh S. Dhanani [4], André M. Comeau [4] & Morgan G. I. Langille[4,6]

Identifying differentially abundant microbes is a common goal of microbiome studies. Multiple methods are used interchangeably for this purpose in the literature. Yet, there are few large-scale studies systematically exploring the appropriateness of using these tools interchangeably, and the scale and significance of the differences between them. Here, we compare the performance of 14 differential abundance testing methods on 38 16S rRNA gene datasets with two sample groups. We test for differences in amplicon sequence variants and operational taxonomic units (ASVs) between these groups. Our findings confirm that these tools identified drastically different numbers and sets of significant ASVs, and that results depend on data pre-processing. For many tools the number of features identified correlate with aspects of the data, such as sample size, sequencing depth, and effect size of community differences. ALDEx2 and ANCOM-II produce the most consistent results across studies and agree best with the intersect of results from different approaches. Nevertheless, we recommend that researchers should use a consensus approach based on multiple differential abundance methods to help ensure robust biological interpretations.

[1] Department of Microbiology and Immunology, Dalhousie University, Halifax, NS, Canada. [2] Department of Mathematics and Statistics, Dalhousie University, Halifax, NS, Canada. [3] Department of Computer Science, Dalhousie University, Halifax, NS, Canada. [4] Integrated Microbiome Resource, Dalhousie University, Halifax, NS, Canada. [5] Department of Civil and Resource Engineering, Dalhousie University, Halifax, NS, Canada. [6] Department of Pharmacology, Dalhousie University, Halifax, NS, Canada. [7] These authors contributed equally: Jacob T. Nearing and Gavin M. Douglas. ✉email: jacob.nearing@dal.ca

Microbial communities are frequently characterized by DNA sequencing. Marker gene sequencing, such as 16S rRNA gene sequencing, is the most common form of microbiome profiling and enables the relative abundances of taxa to be compared across different samples. A frequent and seemingly simple question to investigate with this type of data is: which taxa significantly differ in relative abundance between sample groupings? Newcomers to the microbiome field may be surprised to learn that there is little consensus on how best to approach this question. Indeed, there are numerous ongoing debates regarding the best practices for differential abundance (DA) testing with microbiome data[1,2].

One area of disagreement is whether read count tables should be rarefied (i.e., subsampled) to correct for differing read depths across samples[3]. This approach has been heavily criticized because excluding data could reduce statistical power and introduce biases. In particular, using rarefied count tables for standard tests, such as the *t*-test and Wilcoxon test, can result in unacceptably high false positive rates[4]. Nonetheless, microbiome data is still frequently rarefied because it can simplify analyses, particularly for methods that do not control for variation in read depth across samples. For example, LEfSe[5] is a popular method for identifying differentially abundant taxa that first converts read counts to percentages. Accordingly, read count tables are often rarefied before being input into this tool so that variation in sample read depth does not bias analyses. Without addressing the variation in depth across samples by some approach, the richness can drastically differ between samples due to read depth alone.

A related question to whether data should be rarefied is whether rare taxa should be filtered out. This question arises in many high-throughput datasets, where the burden of correcting for many tests can greatly reduce statistical power. Filtering out potentially uninformative features before running statistical tests can help address this problem, although in some cases this can also have unexpected effects such as increases in false positives[6]. Importantly, this filtering must be independent of the test statistic evaluated (referred to as Independent Filtering). For instance, hard cut-offs for the prevalence and abundance of taxa across samples, and not within one group compared with another, are commonly used to exclude rare taxa[7]. This data filtering could be especially important for microbiome datasets because they are often extremely sparse. Nonetheless, it remains unclear whether filtering rare taxa has much effect on DA results in practice.

Another contentious area is regarding which statistical distributions are most appropriate for analyzing microbiome data. Statistical frameworks based on a range of distributions have been developed for modeling read count data. For example, DESeq2[8] and edgeR[9] are both tools that assume read counts follow a negative binomial distribution. To identify differentially abundant taxa, a null and alternative hypothesis are compared for each taxon. The null hypothesis states that the same setting for certain parameters of the negative binomial solution explain the distribution of taxa across all sample groupings. The alternative hypothesis states that different parameter settings are needed to account for differences between sample groupings. If the null hypothesis can be rejected for a specific taxon, then it is considered differentially abundant. This idea is the foundation of distribution-based DA tests, including other methods such as corncob[10] and metagenomeSeq[11], which model microbiome data with the beta-binomial and zero-inflated Gaussian distributions, respectively.

Finally, it has recently become more widely appreciated that sequencing data are compositional[12], meaning that sequencing only provides information on the relative abundance of features and that each feature's observed abundance is dependent on the observed abundances of all other features. This characteristic means that false inferences are commonly made when standard methods, intended for absolute abundances, are used with taxonomic relative abundances. Compositional data analysis (CoDa) methods circumvent this issue by reframing the focus of analysis to ratios of read counts between different taxa within a sample[13,14]. The difference among CoDa methods considered in this paper is what abundance value is used as the denominator, or the reference, for the transformation. The centered log-ratio (CLR) transformation is a CoDa approach that uses the geometric mean of the read counts of all taxa within a sample as the reference/denominator for that sample. In this approach all taxon read counts within a sample are divided by this geometric mean and the log fold changes in this ratio between samples are compared. An extension of this approach is implemented in the tool ALDEx2[15]. The additive log-ratio transformation is an alternative approach where the reference is the count abundance of a single taxon, which should be present with low variance in read counts across samples. In this case the ratio between the reference taxon chosen (denominator) and each taxon in that sample are compared across different sample groupings. ANCOM is one tool that implements this additive log-ratio approach[16].

Regardless of the above choices, evaluating the numerous options for analyzing microbiome data has proven difficult. This is largely because there are no gold standards to compare DA tool results. Simulating datasets with specific taxa that are differentially abundant is a partial solution to this problem, but it is imperfect. For example, it has been noted that parametric simulations can result in circular arguments for specific tools, making it difficult to assess their true performance[17]. It is unsurprising that distribution-based methods perform best when applied to simulated data based on that distribution. Nonetheless, simulated data with no expected differences has been valuable for evaluating the false discovery rate (FDR) of these methods. Based on this approach it has become clear that many of the methods produce unacceptably high numbers of false positive identifications[3,18,19]. Similarly, based on simulated datasets with spiked taxa it has been shown that these methods can drastically vary in statistical power[17,18].

Although these general observations have been well substantiated, there is less agreement regarding the performance of tools across evaluation studies. Certain observations have been reproducible, such as the higher FDR of edgeR and metagenomeSeq. Similarly, ALDEx2 has been repeatedly shown to have low power to detect differences[17,19]. In contrast, both ANCOM and limma voom[20,21] have been implicated as both accurately and poorly controlling the FDR, depending on the study[3,17,19]. To further complicate comparisons, different sets of tools and dataset types have been analyzed across evaluation studies. This means that, on some occasions, the best performing method in one evaluation is missing from another. In addition, certain popular microbiome-specific methods, such as MaAsLin2[22], have been missing from past evaluations. Finally, many evaluations limit their analysis to a small number of datasets that do not represent the breadth of datasets found in 16S rRNA gene sequencing studies.

Given the inconsistencies across these studies it is important that additional, independent evaluations be performed to elucidate the performance of current DA methods. This is particularly important as these tools are typically used interchangeably in microbiome research. Accordingly, herein we have conducted additional evaluations of common DA tools across 38 two-group 16S rRNA gene datasets. We first present the concordance of the methods on these datasets to investigate how consistently the methods cluster and perform in general, with and without the removal of rare taxa. Next, based on artificially subsampling the datasets into two groups where no differences are expected, we present the observed false positive rate for each DA tool. Lastly, we present an evaluation of how consistent biological

interpretations would be across diarrheal and obesity datasets depending on which tool was applied. Our work enables improved assessment of these DA tools and highlights which key recommendations made by previous studies hold in an independent evaluation. Furthermore, our analysis shows various characteristics of DA tools that authors can use to evaluate published literature within the field.

## Results

**High variable in number of significant ASVs identified**. To investigate how different DA tools impact biological interpretations across microbiome datasets, we tested 14 different DA testing approaches (Table 1) on 38 different microbiome datasets with a total of 9,405 samples. These datasets corresponded to a range of environments, including the human gut, plastisphere, freshwater, marine, soil, wastewater, and built environments (Supplementary Data 1). The features in these datasets corresponded to both ASVs and clustered operational taxonomic units, but we refer to them all as ASVs below for simplicity.

We also investigated how prevalence filtering each dataset prior to analysis impacted the observed results. We chose to either use no prevalence filtering (Fig. 1a) or a 10% prevalence filter that removed any ASVs found in fewer than 10% of samples within each dataset (Fig. 1b).

We found that in both the filtered and unfiltered analyses the percentage of significant ASVs identified by each DA method varied widely across datasets, with means ranging from 3.8–32.5% and 0.8–40.5%, respectively. Interestingly, we found that many tools behaved differently between datasets. Specifically, some tools identified the most features in one dataset while identifying only an intermediate number in other datasets. This was especially evident in the unfiltered datasets (Fig. 1a).

Despite the variability of tool performance between datasets, we did find that several tools tended to identify more significant hits (Supplementary Fig. 1C, D). In the unfiltered datasets, we found that limma voom (TMMwsp; mean: 40.5%; SD: 41% / TMM; mean: 29.7%; SD: 37.5%), Wilcoxon (CLR; mean: 30.7%; SD: 42.3%), LEfSe (mean: 12.6%; SD: 12.3%), and edgeR (mean: 12.4%; SD: 11.4%) tended to find the largest number of significant ASVs compared with other methods. Interestingly, in a few datasets, such as the Human-ASD and Human-OB (2) datasets, edgeR found a higher proportion of significant ASVs than any other tool. In addition, we found that limma voom (TMMwsp) found the majority of ASVs to be significant (73.5%) in the Human-HIV (3) dataset while the other tools found 0–11% ASVs to be significant (Fig. 1a). We found that both limma voom methods identified over 99% of ASVs to be significant in several cases such as the Built-Office and Freshwater-Arctic datasets. This is most likely due to the high sparsity of these datasets causing the tools' reference sample selection method (upper-quartile normalization) to fail. Such extreme findings were also seen in the Wilcoxon (CLR) output, where more than 90% of ASVs were called as significant in eight separate datasets. We found similar, although not as extreme, trends with LEfSe where in some datasets, such as the Human-T1D (1) dataset, the tool found a much higher percentage of significant hits (3.5%) compared with all other tools (0–0.4%). This observation is most likely a result of LEfSe filtering significant features by effect size rather than using FDR correction to reduce the number of false positives. We found that two of the three compositionally aware methods we tested identified fewer significant ASVs than the other tools tested. Specifically, ALDEx2 (mean: 1.4%; SD: 3.4%) and ANCOM-II (mean: 0.8%; SD: 1.8%) identified the fewest significant ASVs. We found the conservative behavior of these tools to be consistent across all 38 datasets we tested.

Overall, the results based on the filtered tables were similar, although there was a smaller range in the number of significant features identified by each tool. All tools except for ALDEx2 found a lower number of total significant features when compared with the unfiltered dataset (Supplementary Fig. 1C, D). As with the unfiltered data, ANCOM-II was the most stringent method (mean: 3.8%; SD: 5.9%), while edgeR (mean: 32.5%; SD: 28.5%), LEfSe (mean: 27.5%; SD: 25.0%), limma voom (TMMwsp; mean: 27.3%; SD: 30.1%/TMM; mean: 23.5%; SD: 27.7%), and Wilcoxon (CLR; mean: 25.4%; SD: 31.7%) tended to output the highest numbers of significant ASVs (Fig. 1b).

To investigate possible factors driving this variation we examined how the number of ASVs identified by each tool correlated with several variables. These variables included dataset richness, variation in sequencing depth between samples, dataset sparsity, and Aitchison's distance effect size (based on PERMANOVA tests). As expected, we found that the number of ASVs identified by all tools positively correlated with the effect size between test groups with Spearman correlation coefficient values ranging between 0.35–0.72 with unfiltered data (Fig. 2a) and 0.31–0.56 for filtered data (Fig. 2b). We also found in the filtered datasets that the number of ASVs found by all tools significantly correlated with the median read depth, range in read depth, and sample size. There was much less consistency in these correlations across the unfiltered data. For instance, only the $t$-test, both Wilcoxon methods, and both limma voom methods correlated significantly with the range in read depth (Fig. 2b). We also found that edgeR was negatively correlated with mean sample richness in the unfiltered analysis. The percentage of ASVs in the unfiltered datasets that were lower than 10% prevalence was also significantly associated with the output of several tools. We also investigated if chimeras could influence the number of significant ASVs detected with results showing very limited impact (Supplementary Fig. 2).

We next investigated whether the significant ASVs identified by the tested DA tools were, on average, at different relative abundances. The clearest outliers were ALDEx2 (median relative abundance of significant ASVs: 0.013%), ANCOM-II (median: 0.024%), and to a lesser degree DESeq2 (median: 0.007%), which tended to find significant features that were at higher relative abundance in the unfiltered datasets (Supplementary Fig. 1A; the medians for all other tools ranged from 0.00004–0.003%). A similar trend for ALDEx2 (median: 0.011%) and ANCOM-II (median: 0.029%) was also apparent in the filtered datasets (Supplementary Fig. 1B; the medians for all other tools ranged from 0.005–0.008%).

Finally, we also examined the discriminatory value of the significant ASVs identified by each tool in the filtered datasets. By discriminatory value we are referring to how well the sample groups can be delineated by a single ASV using hard cut-off abundance values. For this analysis we used either the relative abundances (Supplementary Fig. 3A) or the CLR abundances (Supplementary Fig. 3B) of each significant ASV input into receiver operator curves (ROC) predicting the groups of interest. Raw abundance values were used as input and multiple optimal cut-off points were selected to produce ROCs comparing sensitivity to specificity. We then measured the area under this curve (AUC) of each significant ASV and calculated the mean value across all ASVs identified by each tool. We found that ASVs identified by either ALDEx2 or ANCOM-II had the highest mean AUROC across the tested datasets using both relative abundances and CLR abundances as input (Supplementary Fig. 3). Despite this trend, there were instances where these tools failed to identify any significant ASVs despite other tools achieving relatively high mean AUROCs for the ASVs they identified. For example, in the human-IBD dataset several tools found mean AUROCs of the ASVs they identified ranging from 0.8–0.9 using either CLR or

**Table 1 Differential abundance tools compared in this study.**

| Tool (version) | Input | Norm. | Trans. | Distribution | Covariates | Random effects | Hypothesis test | FDR Corr. | CoDa | Dev. For |
|---|---|---|---|---|---|---|---|---|---|---|
| ALDEx2 (1.18.0) | Counts | None | CLR | Dirichlet-multinomial | Yes* | No | Wilcoxon rank-sum | Yes | Yes | RNA-seq, 16S, MGS |
| ANCOM-II (2.1) | Counts | None | ALR | Non-parametric | Yes | Yes | Wilcoxon rank-sum | Yes | Yes | MGS |
| Corncob (0.1.0) | Counts | None | None | Beta-binomial | Yes | No | Wald (default) | Yes | No | 16S, MGS |
| DESeq2 (1.26.0) | Counts | Modified RLE (default is RLE) | None | Negative binomial | Yes | No | Wald (default) | Yes | No | RNA-seq, 16S, MGS |
| edgeR (3.28.1) | Counts | RLE (default is TMM) | None | Negative binomial | Yes* | No | Exact | Yes | No | RNA-seq |
| LEFse | Rarefied Counts | TSS | None | Non-parametric | Subclass factor only | No | Kruskal–Wallis | No | No | 16S, MGS |
| MaAsLin2 (1.0.0) | Counts | TSS | AST (default is log) | Normal (default) | Yes | Yes | Wald | Yes | No | MGS |
| MaAsLin2 (rare) (1.0.0) | Rarefied counts | TSS | AST (default is log) | Normal (default) | Yes | Yes | Wald | Yes | No | MGS |
| metagenomeSeq (1.28.2) | Counts | CSS | Log | Zero-inflated (log-) Normal | Yes | No | Moderated t | Yes | No | 16S. MGS |
| limma voom (TMM) (3.42.2) | Counts | TMM | Log; Precision weighting | Normal (default) | Yes | Yes | Moderated t | Yes | No | RNA-seq |
| limma voom (TMMwsp) (3.42.2) | Counts | TMMwsp | Log; Precision weighting | Normal (default) | Yes | Yes | Moderated t | Yes | No | RNA-seq |
| t-test (rare) | Rarefied Counts | None | None | Normal | No | No | Welch's t-test | Yes | No | N/A |
| Wilcoxon (CLR) | CLR abundances | None | CLR | Non-parametric | No | No | Wilcoxon rank-sum | Yes | Yes | N/A |
| Wilcoxon (rare) | Rarefied counts | None | None | Non-parametric | No | No | Wilcoxon rank-sum | Yes | No | N/A |

*The tool supports additional covariates if they are provided. ANCOM-II automatically performs ANOVA in this case. ALDEx2 requires that users select the test, and edgeR requires use of a different function (glmFit or glmQLFit instead of exactTest).
ALR additive log-ratio, AST arcsine square-root transformation, CLR centered log-ratio, CoDa compositional data analysis, CSS cumulative sum scaling, FDR Corr. false-discovery rate correction, MGS metagenomic sequencing, RLE relative log expression, TMM trimmed mean of M-values, Trans. transformation, TSS total sum scaling.

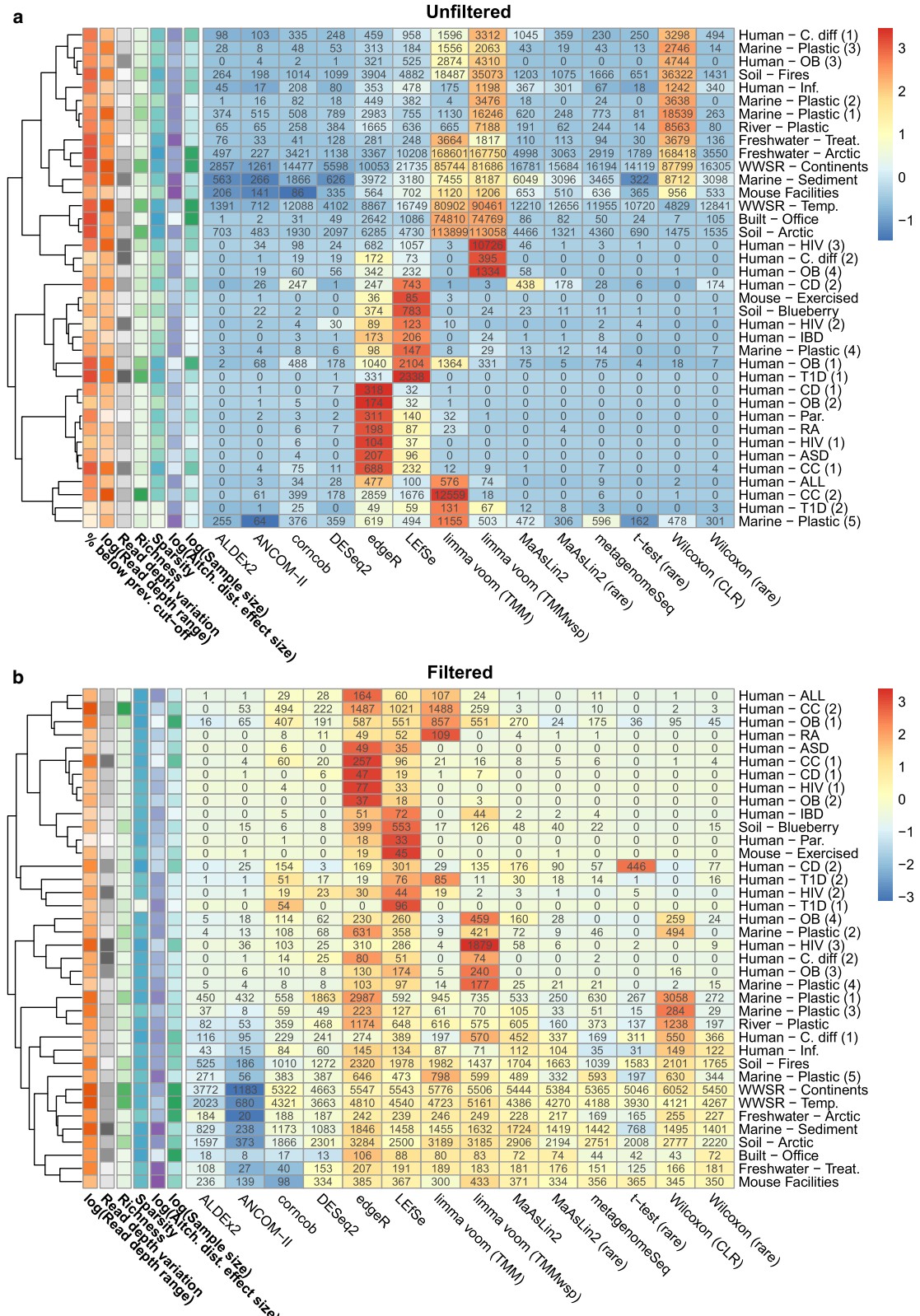

**Fig. 1 Variation in the proportion of significant features depending on the differential abundance method and dataset.** Heatmaps indicate the numbers of significant amplicon sequence variants (ASVs) identified in each dataset by the corresponding tool based on **a** unfiltered data and **b** 10% prevalence-filtered data. Cells are colored based on the standardized (scaled and mean centered) percentage of significant ASVs for each dataset. Additional colored cells in the left-most six columns indicate the dataset characteristics we hypothesized could be driving variation in these results (darker colors indicate higher values). Datasets were hierarchically clustered based on Euclidean distances using the complete method. Abbreviations: prev., previous; TMM, trimmed mean of M-values; TMMwsp, trimmed mean of M-values with singleton pairing; rare, rarefied; CLR, center-log-ratio. Source data are provided as a Source Data file.

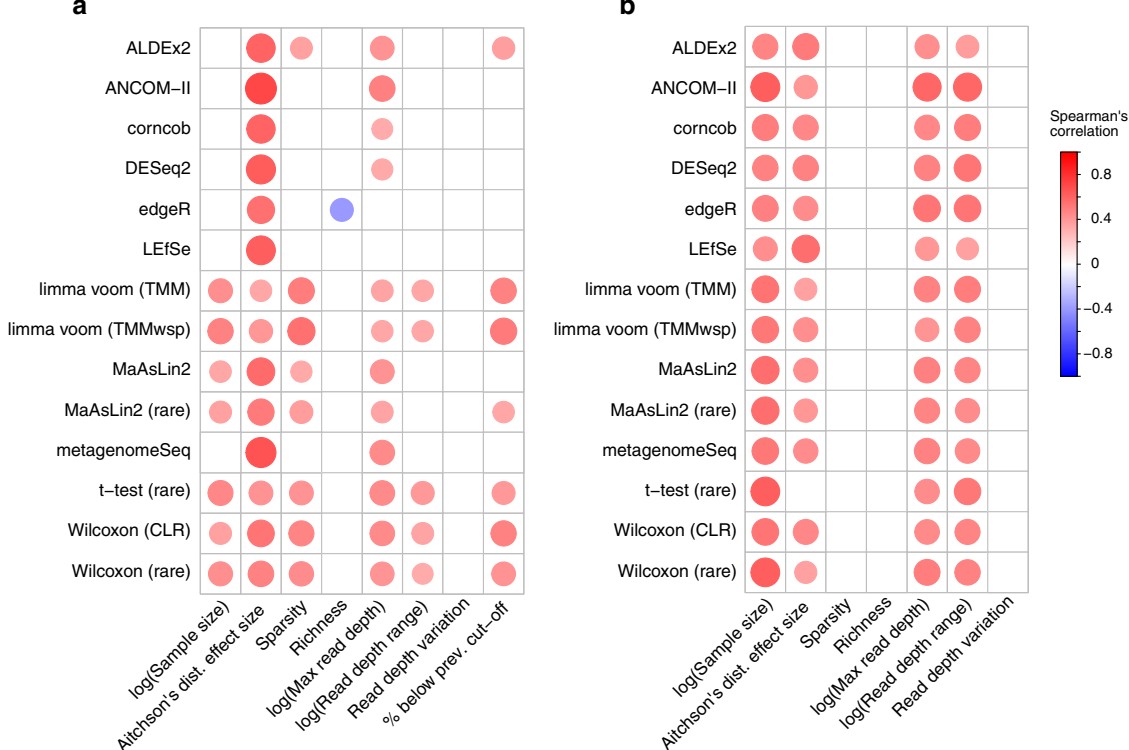

**Fig. 2 Dataset characteristics associated with percentage of significant amplicon sequence variants.** The correlation coefficients (Spearman's rho) are displayed by size and color for the **a** unfiltered and **b** prevalence-filtered data. These correspond to the dataset characteristics correlated with the percentage of significant amplicon sequence variants identified by that tool per dataset. Only significant correlations before multiple comparison correction (*p* < 0.05) are displayed. Abbreviations: prev., previous; TMM, trimmed mean of M-values; TMMwsp, trimmed mean of M-values with singleton pairing; rare, rarefied; CLR, center-log-ratio. Source data are provided as a Source Data file.

relative abundances as input while both ALDEx2 and ANCOM-II failed to identify any significant ASVs.

As the above analysis has the potential to penalize tools that call a higher number of ASVs that are of lower discriminatory values we also investigated the ability of the tested DA methods to identify ASVs above specific AUC thresholds (Supplementary Fig. 4). An important assumption of this analysis is that to accept the exact performance values all ASVs above and below the selected AUROC threshold must be true and false positives, respectively. Although this strict assumption is almost certainly false, it is likely that ASVs above and below the AUC threshold are at least enriched for true and false positives, respectively. We found that at an AUROC threshold of 0.7 both ANCOM-II and ALDEx2 had the highest precision for both relative abundance (medians: 0.99; SD: 0.36 and median: 0.82, SD:0.39) and CLR data (medians: 1.0; SD:0.35 and median: 0.83, SD:0.35). However, they suffered lower recall values based on both relative (medians: 0.17 and 0.02) and CLR-based (medians: 0.06 and 0.01) abundances when compared to the recall scores of tools such as LEfSe and edgeR on relative abundance (median: 0.96 and 0.69) and CLR data (medians: 0.50 and 0.34). When examining CLR data as input we found that limma voom (TMMwsp) had one of the highest F1 scores (median: 0.47) only being outcompeted by the Wilcoxon (CLR) test (mean: 0.70). Examining the data at a higher AUC threshold of 0.9 showed that all tools had relatively high recall scores, apart from some tools such as ANCOM-II, corncob, and *t*-test (rare) on CLR data (medians: 0.5, 0.5, and 0.20). The precision score of all tools at this threshold was low on both relative abundance (range: 0–0.01) and CLR data (range: 0–0.2). This result is unsurprising as in practice we would expect DA tools to identify features that are below a discriminatory threshold of 0.9 AUC.

**High variability of overlapping significant ASVs**. We next investigated the overlap in significant ASVs across tools within each dataset. These analyses provided insight into how similar the interpretations would be depending on which DA method was applied. We hypothesize that tools that produce significant ASVs that highly intersect with the output of other DA tools are the most accurate approaches. Conversely, this may not be the case if tools with similar approaches produce similar sets of significant ASVs and end up identifying the same spurious ASVs. Either way, identifying overlapping significant ASVs across methods provides insights into their comparability.

Based on the unfiltered data, we found that limma voom methods identified similar sets of significant ASVs that were different from those of most other tools (Fig. 3a). However, we also found that many of the ASVs identified by the limma voom methods were also identified as significant based on the Wilcoxon (CLR) approach, despite these being highly methodologically distinct tools. Furthermore, the two Wilcoxon test approaches had highly different consistency profiles, which highlights the impact that CLR-transforming has on downstream results. In contrast, we found that both MaAsLin2 approaches had similar consistency profiles, although the non-rarefied method found slightly lower-ranked features. We also found that the most conservative tools, ALDEx2 and ANCOM-II, primarily identified features that were also identified by almost all other methods. In contrast, edgeR and LEfSe, two tools that often identified the most significant ASVs, output the highest percentage of ASVs that were not identified by any other tool: 11.4% and 9.6%, respectively. Corncob, metagenomeSeq, and DESeq2 identified ASVs at more intermediate consistency profiles.

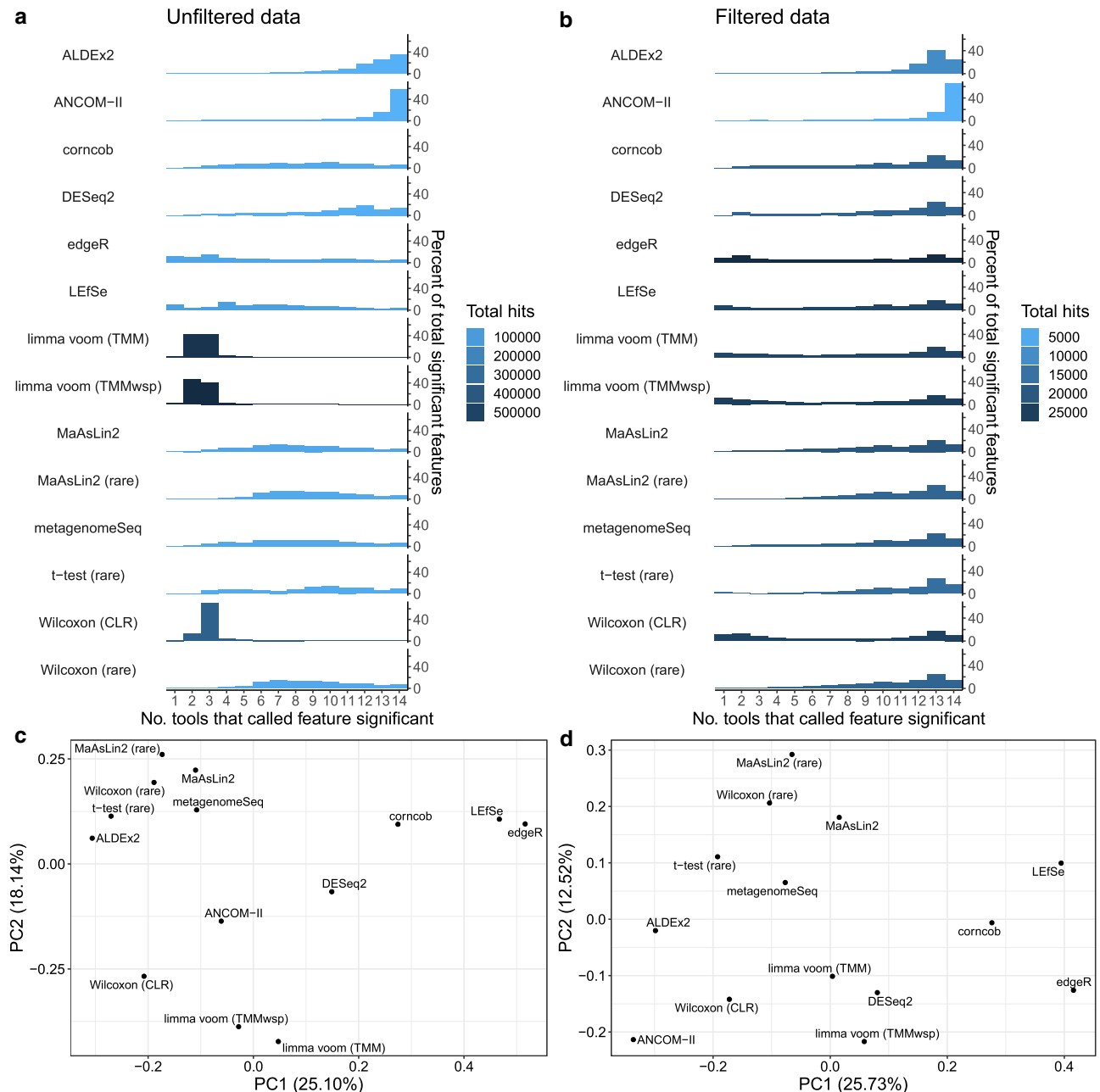

**Fig. 3 Overlap of significant features across tools and tool clustering. a, b** The number of tools that called each feature significant, stratified by features called by each individual tool for the **a** unfiltered and **b** 10% prevalence-filtered data. Results are shown as a percentage of all ASVs identified by each tool. The total number of significant features identified by each tool is indicated by the bar colors. For example, based on the unfiltered data these bars indicate that almost 40% of significant ASVs identified by ALDEx2 were shared across all other tools, while ALDEx2 did not identify any significant ASVs shared by fewer than eight tools. Note that when interpreting these results that they are dependent on which methods were included, and whether they are represented multiple times. For instance, two different workflows for running MaAslin2 are included, which produced similar outputs. **c, d** Plots are displayed for the first two principal coordinates (PCs) for both **c** non-prevalence-filtered and **d** 10% prevalence-filtered data. These plots are based on the mean inter-tool Jaccard distance across the 38 main datasets that we analyzed, computed by averaging over the inter-tool distance matrices for all individual datasets to weight each dataset equally. Abbreviations: TMM, trimmed mean of M-values; TMMwsp, trimmed mean of M-values with singleton pairing; rare, rarefied; CLR, center-log-ratio. Source data are provided as a Source Data file.

The overlap in significant ASVs based on the prevalence-filtered data was similar overall to the unfiltered data results (Fig. 3b). One important exception was that the limma voom approaches identified a much higher proportion of ASVs that were also identified by most other tools, compared with the unfiltered data. Nonetheless, similar to the unfiltered data results, the Wilcoxon (CLR) significant ASVs displayed a bimodal distribution and a strong overlap with limma voom methods.

We also found that overall, the proportion of ASVs consistently identified as significant by more than 12 tools was much higher in the filtered data (mean: 38.6%; SD: 15.8%) compared with the unfiltered data (mean: 17.3%; SD: 22.1%). In contrast with the unfiltered results, corncob, metagenomeSeq, and DESeq2 had lower proportions of ASVs at intermediate consistency ranks. However, ALDEx2 and ANCOM-II once again produced significant ASVs that largely overlapped with most other tools.

A major caveat of the above analysis is that each DA tool produced different numbers of ASVs in total. Accordingly, in principle all of the tools could be identifying the same top ASVs and simply taking varying degrees of risk when identifying less clearly differential ASVs. To investigate this possibility, we identified the overlap between the 20 top-ranked ASVs per dataset (Supplementary Fig. 5), which included non-significant (but relatively highly ranked) ASVs in some cases. The distribution of these ASVs in the 20 top-ranked hits for all tools was similar to that of all significant ASVs described above. For example, in the filtered data we found that, t-test (rare) (mean: 6.2; SD: 3.8), edgeR, (mean: 6.5; SD: 4.6), corncob (mean: 6.0; SD: 4.1), metagenomeSeq (mean: 5.2; SD: 4.7) and DESeq2 (mean: 4.7; SD: 4.5) had the highest number of ASVs only identified by that particular tool as being in the top 20. On average only a small number of ASVs were amongst the top 20 ranked of all tools in both the filtered (mean: 0.21; SD: 0.62) and unfiltered (mean: 0.11; SD: 0.31) datasets (Supplementary Fig. 5). The above analyses summarized the consistency in tool outputs, but it is difficult to discern which tools performed most similarly from these results alone. To identify overall similarly performing tools we conducted principal coordinates analysis based on the Jaccard distance between significant sets of ASVs (Fig. 3c, d). These analyses provide insight into how similar the results of different tools are expected to be, which could be due to methodological similarities between them. However, this does not provide clear evidence for which tools are the most accurate. One clear trend for both unfiltered and filtered data is that edgeR and LEfSe cluster together and are separated from other methods on the first principal coordinate. Interestingly, corncob, which is a methodologically distinct approach, also clusters relatively close to these two methods on the first principal component. This may reflect that the distributions that these two methods rely upon become similar when considering the parameter values often associated with microbiome data.

The major outliers on the second principal coordinate differ depending on whether the data was prevalence-filtered. For the unfiltered data, the main outliers are the limma voom methods, followed by Wilcoxon (CLR; Fig. 3c). In contrast, ANCOM-II is the sole major outlier on the second principal component based on filtered data (Fig. 3d). These visualizations highlight the major tool clusters based on the resulting sets of significant ASVs. However, the percentage of variation explained by the top two components is relatively low in each case, which means that substantial information regarding potential tool clustering is missing from these panels (Supplementary Fig. 6 and Supplementary Fig. 7). For instance, ANCOM-II and corncob are major outliers on the third and fourth principal coordinates, respectively, of the unfiltered data analysis, which highlights the uniqueness of these methods.

**False discovery rate of microbiome differential abundance tools depends on the dataset**. We next evaluated how the DA tools performed in cases where no significant hits were expected. These cases corresponded to sub-samplings of eight of the 38 datasets presented above. For each dataset we selected the most frequently sampled group and within this sample grouping we randomly reassigned them as case or control samples. Each DA tool was then run on this subset of randomly assigned samples and results were compared. Due to the random nature of assignment and the similar composition of samples from the same metadata grouping (e.g. healthy humans) we would expect tools to not identify any ASVs as being differential abundant. Through this approach we were able to infer the false positive characteristics of each tool. In other words, we determined what percentage of tested

ASVs was called as significant by each tool even when there is no difference expected between the sample groups.

The clearest trend for both unfiltered and filtered data is that certain outlier tools have relatively high FDRs in this context while most others identify few false positives (Fig. 4). Both limma voom methods output highly variable percentages of significant ASVs, especially based on the unfiltered data (Fig. 4a). In 5/8 of the unfiltered datasets, the limma voom methods identified more than 5% of ASVs as significant on average due to many high value outliers. Only ALDEx2 and t-test (rare) consistently identified no ASVs as significantly different in the unfiltered data analyses. However, both MaAsLin2 and Wilcoxon (rare) found no significant features in the majority of tested datasets (6/8 and 7/8 respectively). Two clear outliers in the filtered data analyses were edgeR (mean: 0.69–27.9%) and LEfSe (mean: 3.4–5.1%) which consistently identified more significant hits compared with other tools (Fig. 4b). However, it should be noted that in some datasets Corncob, DESeq2 and the limma methods also performed poorly.

Overall, we found that the raw numbers of significant ASVs were lower in the filtered dataset than in the unfiltered data (as expected due to many ASVs being filtered out), and that most tools identified only a small percentage of significant ASVs, regardless of filtering procedure. The exceptions were the two limma voom methods, which had high FDRs with unfiltered data, and edgeR and LEfSe, which had high FDRs on the filtered data. Although these tools stand out on average, we also observed that in several replicates on the unfiltered datasets, the Wilcoxon (CLR) approach identified almost all features as statistically significant (Fig. 4a). This was also true for both limma voom methods, which highlights that a minority of replicates are driving up the average FDR of these methods.

We investigated the outlier replicates for the Wilcoxon (CLR) approach and found that the mean differences in read depth between the two tested groups were consistently higher in replicates in which 30% or more of ASVs were significant (Supplementary Fig. 8). These differences were associated with similar differences in the geometric mean abundances per-sample (i.e., the denominator of the CLR transformation) between the test groups. Specifically, per dataset, these outlier replicates commonly displayed the most extreme mean difference in geometric mean between the test groups and were otherwise amongst the top ten most extreme replicates. Interestingly, the pattern of differential read depth was absent when examining outlier replicates for the limma voom methods (Supplementary Fig. 8).

**Tools vary in consistently across diarrhea case-control datasets**. Separate from the above analysis comparing consistency between tools on the same dataset, we next investigated whether certain tools provide more consistent signals across datasets of the same disease. This analysis focused on the genus-level across tools to help limit inter-study variation. We specifically focused on diarrhea as a phenotype, which has been shown to exhibit a strong effect on the microbiome and to be relatively reproducible across studies[23].

We acquired five datasets for this analysis representing the microbiome of individuals with diarrhea compared with individuals without diarrhea (see Methods). We ran all DA tools on each individual filtered dataset and restricted our analyses to the 218 genera found across all datasets. Like our ASV-level analyses, the tools substantially varied in terms of the number of significant genera identified. For instance, ALDEx2 identified a mean of 17.6 genera as significant in each dataset (SD: 17.4), while edgeR identified a mean of 46.0 significant genera (SD: 12.9). Tools that generally identify more genera as significant are accordingly more likely to identify genera as consistently significant compared with tools with fewer significant hits.

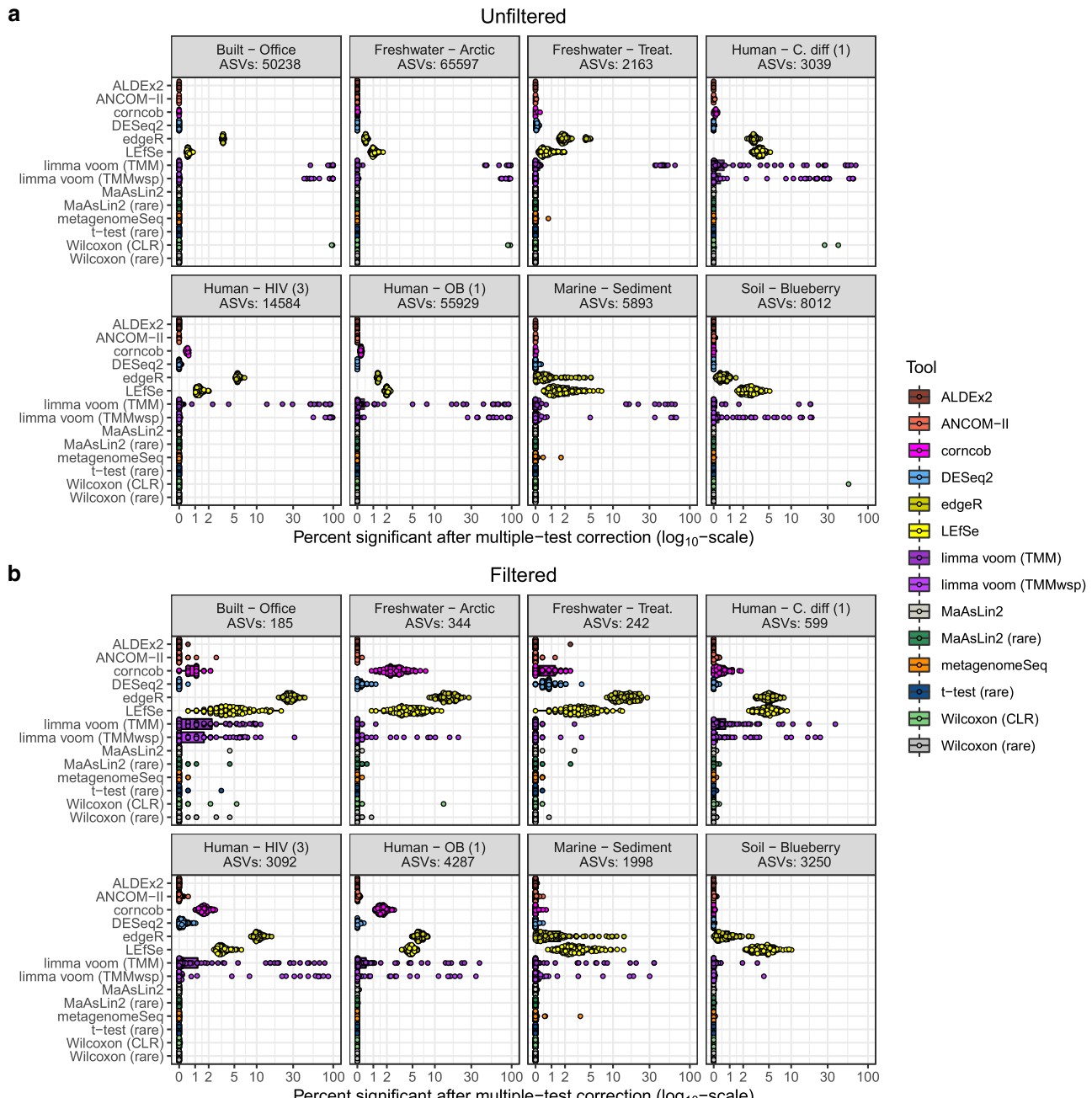

**Fig. 4 Distribution of false discovery rate simulation replicates for both unfiltered and filtered data.** The percentage of amplicon sequence variants that are significant after performing Benjamini–Hochberg correction of the p-values (using a cut-off of 0.05) are shown for each separate dataset and tool. Interquartile range (IQR) of boxplots represent the 25th and 75th percentiles while maxima and minima represent the maximum and minimum values outside 1.5 times the IQR. Notch in the middle of the boxplot represent the median. Note that the x-axis is on a pseudo-log$_{10}$ scale. **a** Represents unfiltered datasets while **b** represents datasets filtered using a 10% prevalence requirement for each ASV. Datasets and tools were run 100 times while randomly assigning samples from the same environment and original groupings to one of two new randomly selected groupings. Differential abundance analysis was then performed on the two random groupings. Note that in the unfiltered datasets 100 replicates was only run 3 of the 8 datasets (Freshwater—Arctic, Soil —Blueberry, Human—OB (1)) with 100 ALDEx2 replications also being run in the Human - HIV (3) dataset. All other unfiltered datasets were run with 10 replicates due to computational limitations. Abbreviations: TMM, trimmed mean of M-values; TMMwsp, trimmed mean of M-values with singleton pairing; rare, rarefied; CLR, center-log-ratio. Source data are provided as a Source Data file.

Accordingly, inter-tool comparisons of the number of times each genus was identified as significant would not be informative.

Instead, we analyzed the observed distribution of the number of studies that each genus was identified as significant in compared with the expected distribution given random data. This approach enabled us to compare the tools based on how much more consistently each tool performed relative to its own random expectation. For instance, on average edgeR identified significant genera more consistently across studies compared with ALDEx2 (mean numbers of datasets that genera were found in across studies were 1.67 and 1.54 for edgeR and ALDEx2, respectively). However, this observation was simply driven by the increased number of significant genera identified by edgeR. Indeed, when compared with the random expectation, ALDEx2

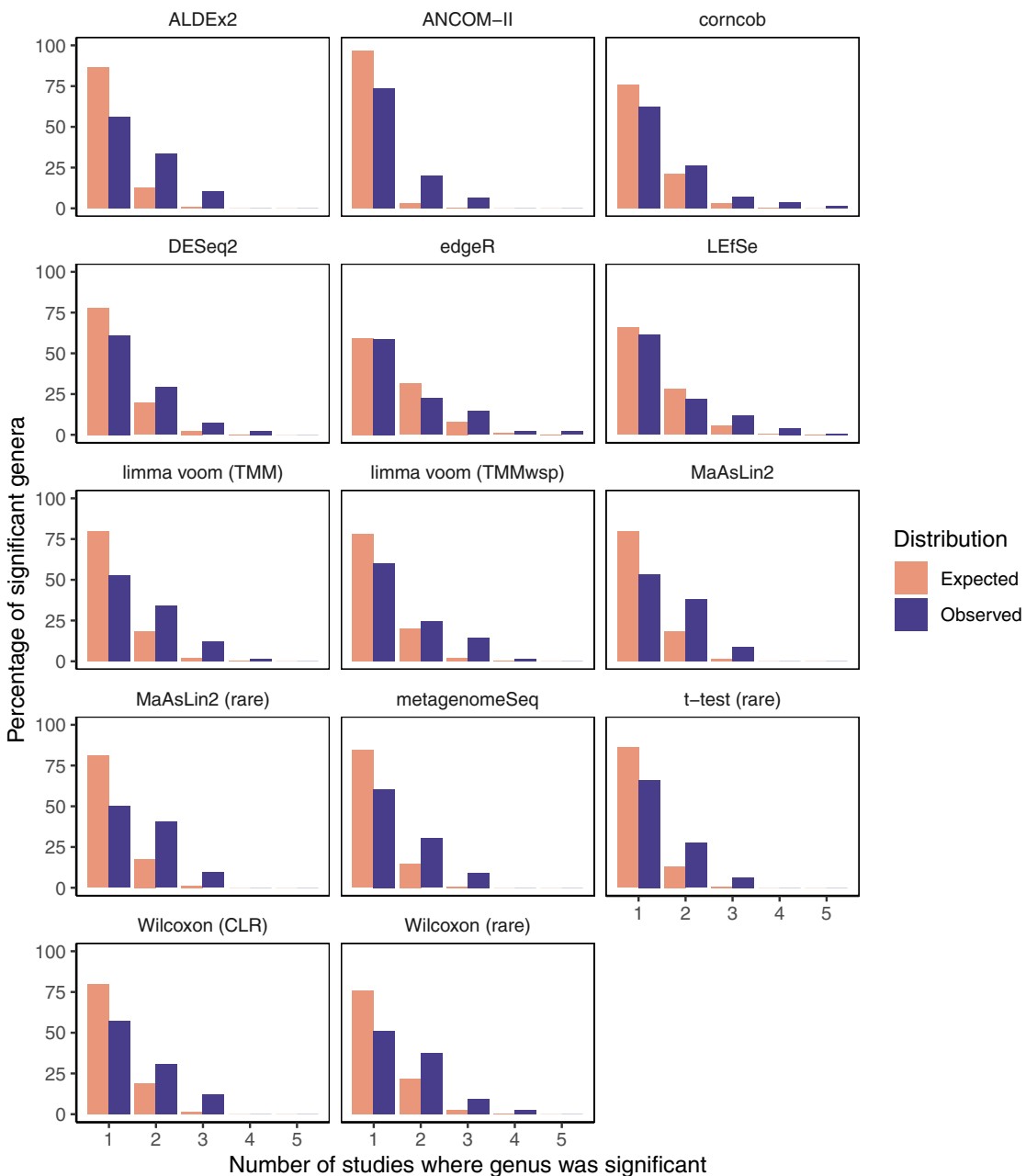

**Fig. 5 Observed consistency of significant genera across diarrhea datasets is higher than the random expectation overall.** These barplots illustrate the distributions of the number of studies for which each genus was identified as significant (excluding genera never found to be significant). The random expectation distribution is based on replicates of randomly selecting genera as significant and then computing the consistency across studies. Abbreviations: TMM, trimmed mean of M-values; TMMwsp, trimmed mean of M-values with singleton pairing; rare, rarefied; CLR, center-log-ratio. Source data are provided as a Source Data file.

displayed a 1.35-fold increase ($p < 0.001$) of consistency in calling significant genera in the observed data. In contrast, edgeR produced results that were only 1.10-fold more consistent compared with the random expectation ($p = 0.002$).

ALDEx2 and edgeR represent the extremes of how consistently tools identify the same genera as significant across studies, but there is a large range (Fig. 5). Notably, all tools were significantly more consistent than the random expectation across these datasets ($p < 0.05$) (Table 2). In addition to ALDEx2, the other top performing approaches based on this evaluation included limma voom (TMM), both MaAsLin2 workflows, and ANCOM-II.

We conducted a similar investigation across five obesity 16S rRNA gene datasets, which was more challenging to interpret due

to the lower consistency in general (Supplementary Table 1). Specifically, most significant genera were called in only a single study and only MaAslin2 (both with non-rarefied and rarefied data), the *t*-test (rare) approach, ALDEx2, and the limma voom (TMMwsp) approach performed significantly better than expected by chance ($p < 0.05$). The MaAsLin2 (rare) approach produced by far the most consistent results based on these datasets (fold difference: 1.23; $p = 0.003$).

## Discussion

Herein we have compared the performance of commonly used DA tools on 16S rRNA gene datasets. While it might be argued

**Table 2 Comparison of observed and expected consistency in differentially abundant genera across five diarrhea datasets.**

| Tool | No. sig. genera | Max overlap | Mean exp. | Mean obs. | Fold diff. | p |
|---|---|---|---|---|---|---|
| ALDEx2 | 57 | 3 | 1.141 | 1.544 | 1.353 | <0.001 |
| limma voom (TMM) | 76 | 4 | 1.22 | 1.618 | 1.326 | <0.001 |
| MaAsLin2 (rare) | 74 | 3 | 1.204 | 1.595 | 1.325 | <0.001 |
| ANCOM-II | 15 | 3 | 1.033 | 1.333 | 1.29 | <0.001 |
| MaAsLin2 | 79 | 3 | 1.215 | 1.557 | 1.281 | <0.001 |
| Wilcoxon (rare) | 88 | 4 | 1.269 | 1.625 | 1.281 | <0.001 |
| metagenomeSeq | 66 | 3 | 1.164 | 1.485 | 1.276 | <0.001 |
| Wilcoxon (CLR) | 82 | 3 | 1.22 | 1.549 | 1.27 | <0.001 |
| limma voom (TMMwsp) | 85 | 4 | 1.239 | 1.565 | 1.263 | <0.001 |
| t-test (rare) | 62 | 3 | 1.145 | 1.403 | 1.225 | <0.001 |
| corncob | 87 | 5 | 1.275 | 1.552 | 1.217 | <0.001 |
| DESeq2 | 82 | 4 | 1.246 | 1.512 | 1.213 | <0.001 |
| LEfSe | 119 | 5 | 1.408 | 1.613 | 1.146 | <0.001 |
| edgeR | 138 | 5 | 1.509 | 1.667 | 1.105 | 0.002 |

No. sig. genera: Number of genera significant in at least one dataset; Max overlap: Maximum number of datasets where a genus was called significant by this tool; Mean exp.: Mean number of datasets that each genus is expected to be significant in (of the genera that are significant at least once); Mean obs.: Mean number of datasets that each genus was observed to be significant in (of the genera that are significant at least once); Fold diff.: Fold difference of mean observed over mean expected number of times significant genera are found across multiple datasets; p: p-value based on one-tailed permutation test that used the 'Mean obs.' as the test statistic. Note that <0.001 is indicated instead of exact values, because 0.001 was the minimum non-zero p-value we could estimate based on our permutation approach. Source data are provided as a Source Data file.

that differences in tool outputs are expected given that they test different hypotheses, we believe this perspective ignores how these tools are used in practice. In particular, these tools are frequently used interchangeably in the microbiome literature. Accordingly, an improved understanding of the variation in DA method performance is crucial to properly interpret microbiome studies. We have illustrated here that these tools can produce substantially different results, which highlights that many biological interpretations based on microbiome data analysis are likely not robust to DA tool choice. These results might partially account for the common observation that significant microbial features reported in one dataset only marginally overlap with significant hits in similar datasets. However, it should be noted that this could also be due to numerous other biases that affect microbiome studies[24]. Our findings should serve as a cautionary tale for researchers conducting their own microbiome data analysis and reinforce the need to accurately report the findings of a representative set of different analysis options to ensure robust results are reported. Importantly, readers should not misinterpret our results to mean that 16S rRNA gene data is less reliable than other microbiome data types, such as shotgun metagenomics and metabolomics. We expect similar issues are affecting analyses with these data types, as they have similar or even higher levels of sparsity and inter-sample variation. Nonetheless, despite the high variation across DA tool results, we were able to characterize several consistent patterns produced by various tools that researchers should keep in mind when assessing both their own results and results from published work.

Two major groups of DA tools could be distinguished by how many significant ASVs they tended to identify. We found that limma voom, edgeR, Wilcoxon (CLR), and LEfSe output a high number of significant ASVs on average. In contrast, ALDEx2 and ANCOM-II tended to identify only a relatively small number of ASVs as significant. We hypothesize that these latter tools are more conservative and have higher precision, but with a concomitant probable loss in sensitivity. This hypothesis is related to our observation that significant ASVs identified by these two tools tended to also be identified by almost all other differential abundance methods, which we interpret to be ASVs that are more likely to be true positives. Furthermore, it was clear that in most, but not all, cases these two methods tended to identify the most discriminatory ASVs that were found by other tools. We believe that the lower number of ASVs identified by these approaches

could be due to multiple reasons. ALDEx2's conservative nature is most likely due to its Monte Carlo Dirichlet sampling approach which down weights low abundance ASVs. On the other hand, ANCOM-II's conservative nature could be attributed to its large multiple testing burden. Furthermore, the variance in the number of features identified could also be attributed to pre-processing steps several tools use to remove potential ASVs for testing. This includes corncob that does not report significance values for ASVs only found in one group or ANCOM-II's ability to remove structural zeros. While it is unclear why some methods have much higher significance rates than other tools it could be due to several reasons. These include LEfSe's choice not to correct significance values for false discovery or Wilcoxon (CLR)'s inability to consider differences in sequencing depth between metadata groupings. It should be noted that in some cases authors haven't chosen to apply FDR p-value correction to LEfSe output, when not including a subclass, however, this is not the default behavior of this tool[25].

Given that ASVs commonly identified as significant using a wide range of approaches are likely more reliable, it is noteworthy that significant ASVs in the unfiltered data tended to be called by fewer tools. This was particularly evident for both limma voom approaches and the Wilcoxon (CLR) approach. Although it is possible that many of these significant ASVs are incorrectly missed by other tools, it is more likely that these tools are simply performing especially poorly on unfiltered data due to several reasons, such as data sparsity.

This issue with the limma voom approaches was also highlighted by high false positive rates on several unfiltered randomized datasets, which agrees with a past FDR assessment of this approach[17]. We believe that this issue is most likely driven by the inability for TMM normalization methods to deal with highly sparse datasets as filtering the data resulted in performance more in line with other DA methods[9]. It is important to acknowledge that our randomized approach for estimating FDR is not a perfect representation of real data; that is, real sample groupings will likely contain some systematic differences in microbial abundances—although the effect size may be very small—whereas our randomized datasets should have none. Accordingly, identifying only a few significant ASVs under this approach is not necessarily proof that a tool has a low FDR in practice. However, tools that identified many significant ASVs in the absence of distinguishing signals likely also have high FDR on real data.

Two additional problematic tools based on this analysis were edgeR and LEfSe. The edgeR method has been previously found to exhibit a high FDR on several occasions[17,18] Although metagenomeSeq also has been flagged as such[17,18], that was not the case in our analysis. This agrees with a recent report that metagenomeSeq (using the zero-inflated log-normal approach, as we did) appropriately controlled the FDR, but exhibited low power[26]. There have been mixed results previously regarding whether ANCOM appropriately controls the FDR[3,17], but the results from our limited analysis suggest that this method is conservative and controls the FDR while sometimes potentially missing true positives, as evident in our discriminatory analysis.

Related to this point, we found that ANCOM-II performed better than average at identifying the same genera as significantly DA across five diarrhea-related datasets despite only identifying a mean of four genera as significant per dataset. Nonetheless, the ANCOM-II results were less consistent than ALDEx2, both MaAsLin2 workflows, and limma voom (TMM). The tools that produced the least consistent results across datasets (relative to the random expectation) included the *t*-test (rare) approach, LEfSe, and edgeR. The random expectation in this case was quite simplistic; it was generated based on the assumption that all genera were equally likely to be significant by chance. This assumption must be invalid to some degree simply because some genera are more prevalent than others across samples. Accordingly, it is surprising that the tools produced only marginally more consistent results than expected.

Although this cross-data consistency analysis was informative, it was interesting to note that not all environments and datasets are appropriate for this comparison. Specifically, we found that the consistency of significant genera across five datasets comparing obese and control individuals was no higher than expected by chance for most tools. This observation does not necessarily reflect that there are few consistent genera that differ between obese and non-obese individuals; it could instead simply reflect technical and/or biological factors that differ between the particular datasets we analyzed[1]. Despite these complicating factors, it is noteworthy that the MaAsLin2 workflows and ALDEx2 produced more consistent results than expected based on these datasets.

We believe the above observations regarding DA tools are valuable, but many readers are likely primarily interested in hearing specific recommendations. Indeed, the need for standardized practices in microbiome analysis have recently become better appreciated[27]. One goal of our work was to validate the recommendations of another recent DA method evaluation paper, which found that limma voom, corncob, and DESeq2 performed best overall of the tools they tested[19]. Based on our results, we do not recommend these tools as the sole methods used for data analysis, and instead would suggest that researchers use more conservative methods such as ALDEx2 and ANCOM-II. Although these methods have lower statistical power[17,19], we believe this is an acceptable trade-off given the higher cost of identifying false positives as differentially abundant. However, MaAsLin2 (particularly with rarefied data) could also be a reasonable choice for users looking for increased statistical power at the potential cost of more false positives. We can clearly recommend that users avoid using edgeR (a tool primarily intended for RNA-seq data) as well as LEfSe (without *p*-value correction) for conducting DA testing with 16S rRNA gene data. Users should also be aware that limma voom and the Wilcoxon (CLR) approaches may perform poorly on unfiltered data that is highly sparse. This is particularly true for the Wilcoxon (CLR) approach when read depths greatly differ between groups of interest.

More generally, we recommend that users employ several methods and focus on significant features identified by most tools, while keeping in mind the characteristics of the tools presented within this manuscript. For example, authors may want to present identified taxonomic markers in categories based on the tool characteristics presented within this paper or the number of tools that agree upon its identification. Importantly, applying multiple DA tools to the same dataset should be reported explicitly. Clearly this approach would make results more difficult to biologically interpret, but it would provide a clearer perspective on which differentially abundant features are robust to reasonable changes in the analysis.

A common counterargument to using consensus approaches with DA tools is that there is no assurance that the intersection of the tool outputs is more reliable; it is possible that the tools are simply picking up the same noise as significant. Although we think this is unlikely, in any case running multiple DA tools is still important to give context to reporting significant features. For example, researchers might be using a tool that produces highly non-overlapping sets of significant features compared with other DA approaches. Even if the researchers are confident in their approach, these discrepancies should be made clear when the results are summarized. This is crucial for providing accurate insight into how robust specific findings are expected to be across independent studies, which often use different DA approaches. Similarly, if researchers are most interested in determining if signals from a specific study are reproducible, then they should ensure that they use the same DA approach to help make their results more comparable.

How and whether to conduct independent filtering of data prior to conducting DA tests are other important open questions regarding microbiome data analysis[7]. Although statistical arguments regarding the validity of independent filtering are beyond the scope of this work, intuitively it is reasonable to exclude features found in only a small number of samples (regardless of which groups those samples are in). The basic reason for this is that otherwise the burden of multiple-test correction becomes so great as to nearly prohibit identifying any differentially abundant features. Despite this drawback, many tools identified large numbers of significant ASVs in the unfiltered data. However, these significant ASVs tended to be more tool-specific in the unfiltered data and there was much more variation in the percentage of significant ASVs across tools. Accordingly, we would suggest performing prevalence filtering (e.g., at 10%) of features prior to DA testing, although we acknowledge that more work is needed to estimate an optimal cut-off rather than just arbitrarily selecting one[4].

Another common question is whether DA tools that require input data to be rarefied should be avoided. It is possible that the question of whether to rarefy data has received disproportionate attention in the microbiome field: there are numerous other factors affecting an analysis pipeline that likely affect results more. Indeed, tools that took in rarefied data in our analyses did not perform substantially worse than other methods on average. More specifically, the most consistent inter-tool methods, ANCOM-II and ALDEx2, are based on non-rarefied data, but MaAsLin2 based on rarefied data produced the most consistent results across datasets of the same phenotype. Accordingly, we cannot definitively conclude that DA tools that require input data to be rarefied are less reliable in general. It should be noted that we are referring only to rarefying in the context of DA testing: whether rarefying is advisable for other analyses, such as prior to computing diversity metrics, is beyond the scope of this work[3,4].

Others have investigated the above questions by applying simulations to various DA methods, which has yielded valuable insights[3,17–19]. However, we believe this does not provide the full picture of how these DA tools perform. This is because it has been highlighted that in many scenarios simulations can led to circular arguments where tools that are designed around specific

parameters perform favorably on simulations using those parameters[17]. Without better knowledge of the range of data structures that microbiome sequencing can result in these types of simulation analyses can be difficult to interpret. As such we believe that it was important to test these methods on a wide range of different real-world datasets in order to gain an understanding of how they differed from one another. By doing so we have highlighted the issues of using these tools interchangeably within the literature. Indeed, the question 'which taxa significantly differ in relative abundance between sample groupings?' may be too simple and need further parameterization before it can be answered. This includes information such as what type of abundance the authors are comparing and the tools they plan to use within their analysis. Unfortunately, the variation across tools implies that biological interpretations based on these questions will often drastically differ depending on which DA tool is considered.

In conclusion, the high variation in the output of DA tools across numerous 16S rRNA gene sequencing datasets highlights an alarming reproducibility crisis facing microbiome researchers. While we cannot make a direct simple recommendation of a specific tool based on our analysis, we have highlighted several issues that authors should be aware of while interpreting DA results. This includes that several tools have inappropriately high false discovery rates such as edgeR and LEfSe and as such should be avoided when possible. It is also clear from our analysis that some tools designed for RNA-seq such as limma voom methods cannot deal with the much higher sparsity of microbiome data without including a data filtration step. We have also highlighted that these tools can significantly differ in the number of ASVs that they identify as being significantly different and that some tools are more consistent across datasets than others. Overall, we recommend that authors use the same tools when comparing results between specific studies and otherwise use a consensus approach based on several DA tools to help ensure results are robust to DA choice.

## Methods

**Dataset processing**. Thirty-eight different datasets were included in our main analyses for assessing the characteristics of microbiome differential abundance tools. Three additional datasets were also included for a comparison of differential abundance consistency across diarrhea and obesity-related microbiome datasets. All datasets presented herein have been previously published or are publicly available[23,28–64] (Supplementary Data 1). Most datasets were already available in table format with ASV or operational taxonomic unit abundances while a minority needed to be processed from raw sequences. These raw sequences were processed with QIIME 2 version 2019.7[65] based on the Microbiome Helper standard operating procedure[66]. Primers were removed using cutadapt[67] and stitched together using the QIIME 2 VSEARCH[68] join-pairs plugin. Stitched reads were then quality filtered using the quality-filter plugin and reads were denoised using Deblur[69] to produce amplicon sequence variants (ASVs). Abundance tables of ASVs for each sample were then output into tab-delimited files. Rarefied tables were also produced for each dataset, where the rarefied read depth was taken to be the lowest read depth of any sample in the dataset over 2000 reads (with samples below this threshold discarded).

Chimeric ASVs were identified with the UCHIME2 and UCHIME3 chimera-checking algorithms[70] implemented in VSEARCH (v2.17.1)[68]. Both the UCHIME2 and UCHIME3 de novo approaches were applied in addition to the UCHIME2 reference-based chimera checking approach. For this latter approach we used the SILVA v138.1 short-subunit reference database[71]. We used the default options when running these algorithms.

**Differential abundance testing**. We created a custom shell script (run_all_-tools.sh) that ran each differential abundance tool on each dataset within this study. As input the script took a tab-delimited ASV abundance table, a rarefied version of that same table, and a metadata file that contained a column that split the samples into two groups for testing. This script also accepted a prevalence cut-off filter to remove ASVs below a minimum cut-off, which was set to 10% (i.e., ASVs found in fewer than 10% of samples were removed) for the filtered data analyses we present. Note that in a minority of cases a genus abundance table was input instead, in which case all options were kept the same. When the prevalence

filter option was set, the script also generated new filtered rarefied tables based on an input rarefaction depth.

Following these steps, each individual differential abundance method was run on the input data using either the rarefied or non-rarefied table, depending on which is recommended for that tool. Rarefaction was performed using GUniFrac (version 1.1)[72]. The workflow used to run each differential abundance tool (with run_all_tools.sh) is described below. The first step in each of these workflows was to read the dataset tables into R (version 3.6.3) with a custom script and then ensure that samples within the metadata and feature abundance tables were in the same order. An alpha-value of 0.05 was chosen as our significance cutoff and FDR adjusted $p$-values (using Benjamini-Hochberg adjustment) were used for methods that output $p$-values (with the exception of LEfSe which does not output all $p$-values by default)[73].

**ALDEx2**. We passed the non-rarefied feature table and the corresponding sample metadata to the *aldex* function from the ALDEx2 R package (version 1.18.0)[15] which generated Monte Carlo samples of Dirichlet distributions for each sample, using a uniform prior, performed CLR transformation of each realization, and then performed Wilcoxon tests on the transformed realizations. The function then returned the expected Benjamini-Hochberg (BH) FDR-corrected $p$-value for each feature based on the results the different across Monte Carlo samples.

**ANCOM-II**. We ran the non-rarefied feature table through the R ANCOM-II[16,74] (https://github.com/FrederickHuangLin/ANCOM) (version 2.1) function fea-ture_table_pre_process, which first examined the abundance table to identify outlier zeros and structural zeros[74]. The following packages were imported by ANCOM-II: exactRankTests (version 0.8.31), nlme (version 3.1.149), dplyr (version 0.8.5), ggplot2 (version 3.3.0) and compositions (version 1.40.2). Outlier zeros, identified by finding outliers in the distribution of taxon counts within each sample grouping, were ignored during differential abundance analysis, and replaced with NA. Structural zeros, taxa that were absent in one grouping but present in the other, were ignored during data analysis and automatically called as differentially abundant. A pseudo count of 1 was then applied across the dataset to allow for log transformation. Using the main function ANCOM, all additive log-ratios for each taxon were then tested for significance using Wilcoxon rank-sum tests, and $p$-values were FDR-corrected using the BH method. ANCOM-II then applied a detection threshold as described in the original paper[16], whereby a taxon was called as DA if the number of corrected $p$-values reaching nominal significance for that taxon was greater than 90% of the maximum possible number of significant comparisons.

**corncob**. We converted the metadata and non-rarefied feature tables into a phy-loseq object (version 1.29.0)[75], which we input to corncob's differentialTest function (version 0.1.0)[10]. This function fits each taxon count abundance to a beta-binomial model, using logit link functions for both the mean and overdispersion. Because corncob models each of these simultaneously and performs both differential abundance and differential variability testing[10], we set the null over-dispersion model to be the same as the non-null model so that only taxa having differential abundances were identified. Finally, the function performed significance testing, for which we chose Wald tests (with the default non-bootstrap setting), and we obtained BH FDR-corrected $p$-values as output.

**DESeq2**. We first passed the non-rarefied feature tables to the DESeq function (version 1.26.0)[8] with default settings, except that instead of the default relative log expression (also known as the median-of-ratios method) the estimation of size factors was set to use 'poscounts', which calculates a modified relative log expression that helps account for features missing in at least one sample. The function performed three steps: (1) estimation of size factors, which are used to normalize library sizes in a model-based fashion; (2) estimation of dispersions from the negative binomial likelihood for each feature, and subsequent shrinkage of each dispersion estimate towards the parametric (default) trendline by empirical Bayes; (3) fitting each feature to the specified class groupings with negative binomial generalized linear models and performing hypothesis testing, for which we chose the default Wald test. Finally, using the results function, we obtained the resulting BH FDR-corrected $p$-values.

**edgeR**. Using the phyloseq_to_edgeR function (https://joey711.github.io/phyloseq-extensions/edgeR.html), we added a pseudocount of 1 to the non-rarefied feature table and used the function calcNormFactors from the edgeR R package (version 3.28.1)[9] to compute relative log expression normalization factors. Negative binomial dispersion parameters were then estimated using the functions estimate-CommonDisp followed by estimateTagwiseDisp to shrink feature-wise dispersion estimates through an empirical Bayes approach. We then used the exactTest for negative binomial data[9] to identify features that differ between the specified groups. The resulting $p$-values were then corrected for multiple testing with the BH method with the function topTags.

**LEfSe**. The rarefied feature table was first converted into LEfSe format using the LEfSe script format_input.py[5]. We then ran LEfSe on the formatted table using the run_lefse.py script with default settings and no subclass specifications. Briefly, this command first normalized the data using total sum scaling, which divides each feature count by the total library size. Then it performed a Kruskal-Wallis (which in our two-group case reduces to the Wilcoxon rank-sum) hypothesis test to identify potential differentially abundant features, followed by linear discriminant analysis (LDA) of class labels on abundances to estimate the effect sizes for significant features. From these, only those features with scaled LDA analysis scores above the threshold score of 2.0 (default) were called as differentially abundant. This key step is what distinguished LEfSe from the Wilcoxon test approach based on relative abundances that we also ran. In addition, no multiple-test correction was performed on the raw LEfSe output as only the p-values of significant features above-threshold LDA scores are returned by this tool.

**limma voom**. We first normalized the non-rarefied feature table using the edgeR calcNormFactors function, with either the trimmed mean of M-values (TMM) or TMM with singleton pairing (TMMwsp) option. We choose to run this tool with two different normalization functions as we found the standard TMM normalization technique to struggle with highly spare datasets despite it previously being shown to perform preferentially in DA testing. Furthermore, the TMMwsp method is highlighted within the edgeR package as an alternative for highly sparse data. During either of these normalization steps a single sample was chosen to be a reference sample using upper-quartile normalization. This step failed in some highly sparse abundance tables; in these cases, we instead chose the sample with the largest sum of square-root transformed feature abundances to be the reference sample. After normalization, we used the limma R package (version 3.42.2)[21] function voom to convert normalized counts to log$_2$-counts-per-million and assign precision weights to each observation based on the mean-variance trend. We then used the functions lmFit, eBayes, and topTable in the limma R package to fit weighted linear regression models, perform tests based on an empirical Bayes moderated t-statistic[76] and obtain BH FDR-corrected p-values.

**MaAsLin2**. We entered either a rarefied or non-rarefied feature table into the main Maaslin2 function within the MaAsLin2 R package (version 0.99.12)[22]. We specified arcsine square-root transformation as in the package vignette (instead of the default log) and total sum scaling normalization. For consistency with other tools, we specified no random effects and turned off default standardization. The function fit a linear model to each feature's transformed abundance on the specified sample grouping, tested significance using a Wald test, and output BH FDR-corrected p-values.

**metagenomeSeq**. We first entered the counts and sample information to the function newMRexperiment from the metagenomeSeq R package (version 1.28.2)[11]. Next, we used cumNormStat and cumNorm to apply cumulative sum-scaling normalization, which attempts to normalize sequence counts based on the lower-quartile abundance of features. We then used fitFeatureModel to fit normalized feature counts with zero-inflated log-normal models (with pseudo-counts of 1 added prior to log$_2$ transformation) and perform empirical Bayes moderated t-tests, and MRfulltable to obtain BH FDR-corrected p-values.

**t-test**. We applied total sum scaling normalization to the rarefied feature table and then performed an unpaired Welch's t-test for each feature to compare the specified groups. We corrected the resulting p-values for multiple testing with the BH method.

**Wilcoxon test**. Using raw feature abundances in the rarefied case, and CLR-transformed abundances (after applying a pseudocount of 1) in the non-rarefied case, we performed Wilcoxon rank-sum tests for each feature to compare the specified sample groupings. We corrected the resulting p-values with the BH method.

**Comparing numbers of significant hits between tools**. We compared the number of significant ASVs each tool identified in 38 different datasets. Each tool was run as described above using default settings with some modifications suggested by the tool authors, as noted above. A heatmap representing the number of significant hits found by each tool was constructed using the pheatmap R package (version 1.0.12)[77]. Spearman correlations between the percent of significant ASVs identified by a tool and the following dataset characteristics were computed using the cor.test function in R: sample size, Aitchison's distance effect size as computed using a PERMANOVA test (adonis; vegan, version 2.5.6)[78], sparsity, mean sample ASV richness, median sample read depth, read depth range between samples and the coefficient of variation for read depth within a dataset. In addition, for the unfiltered analyses, we also computed Spearman correlations with the percent of ASVs below 10% prevalence in each dataset (i.e., the percent of ASVs that would be removed to produce the filtered datasets). Correlations were displayed using the R package corrplot (version 0.85) and gridExtra (version 2.3). Dataset manipulation for plotting and reshaping were conducting using the following R packages: doMC

(version 1.3.5), doParallel (version 1.0.15), matrixStats (version 0.56.0), reshape2 (version 1.4.4), plyr (version 1.8.6) and tidyverse (version 1.3.0).

**Cross-tool, within-study differential abundance consistency analysis**. We compared the consistency between different tools within all datasets by pooling all ASVs identified as being significant by at least one tool in the 38 different datasets. The number of methods that identified each ASV as differentially abundant were then tallied. A second way of examining the between method consistency without choosing a specific significance threshold was to examine the overlap between the top 20 ASVs identified by each DA method. To do this ASVs were ranked for each DA method depending on their significance value apart from ANCOM-II, where its W statistic was used for ranking. Like the above analysis, we than tallied the number of methods that identified each ASV as being in its top 20 most differentially abundant ASVs. Multi-panel figures were combined using the R package cowplot (version 1.0.0) and ggplotify (version 0.0.5). To get another view of the data principal coordinate analysis plots were constructed using the mean inter-tool Jaccard distance across the 38 main datasets. Distances were computed by averaging over the inter-tool distance matrices for all individual datasets to weight each dataset equally using the R packages vegan (version 2.5.6) and parallelDist (version 0.2.4). Labels were displayed using the R package ggrepel (version 0.8.1).

**False positive analysis**. To evaluate the false positive rates of each DA method, eight datasets were selected for analysis based on having the largest sample sizes, while also being from diverse environment types. In each dataset, only the most frequent sample group was chosen for analysis to help ensure similar composition among samples tested. Within this grouping, random labels of either case or control were assigned to samples and the various differential abundance methods were tested on them. This was replicated 100 times for each dataset and tool combination aside from ALDEx2, ANCOM-II, and Corncob. These were run using 100 replicates in only 3 of the 8 datasets (Freshwater – Arctic, Soil – Blueberry, Human - OB (1)) with 100 ALDEx2 replications also being run in the Human - HIV (3) dataset. This was due to the long computational time required to run these tools on all datasets. The remaining datasets were replicated 10 times for each of these three tools. After this analysis was completed, the number of differentially abundant ASVs identified by each tool was assessed at an alpha value of 0.05. Boxplots for this data was constructed using the R packages ggplot2 (version 3.3.0), ggbeeswarm (version 0.6.0), and scales (version 1.1.0).

**Cross-study differential abundance consistency analysis**. For this analysis we acquired two additional pre-processed datasets that were not used for other analyses, which are the GEMS1[79] and the dia_schneider[55], datasets (Supplementary Data 1). The processed data for these datasets was acquired from the MicrobiomeDB[48] and the microbiomeHD[23] databases, respectively. These datasets were combined with three of the datasets used elsewhere in this manuscript (Human – C. diff [1 and 2] and Human – Inf.), to bring the number of diarrhea-related datasets to five. These three pre-existing datasets all related to enteric infections that had all been previously demonstrated to show a distinct signal of microbial differences driven by diarrhea in patient samples[23].

For the obesity cross-study analysis we leveraged four datasets that were part of the core 38 datasets: Human – OB (1–4)[37,52,59,64]. We also included an additional obesity dataset, ob_zupancic[80], that we acquired from the microbiomeHD database.

The ASVs in each of these datasets were previously taxonomically classified and so we used these classifications to collapse all feature abundances to the genus level. Note that taxonomic classification was performed using several different methods, which represents another source of technical variation. We excluded unclassified and sensu stricto-labeled genus levels. We then ran all differential abundance tools on these datasets at the genus level. These comparisons were between the diarrhea and non-diarrhea sample groups. The same processing workflow was used for the supplementary obesity dataset comparison as well.

For each tool and study combination, we determined which genera were significantly different at an alpha of 0.05 (where relevant). For each tool we then tallied the number of times each genus was significant, i.e., how many datasets each genus was significant in based on a given tool. The null expectation distributions of these counts per tool were generated by randomly sampling genera from each dataset. The probability of sampling a genus (i.e., calling it significant) was set to be equal to the proportion of actual significant genera. This procedure was repeated 1000 times, with genus replicates equal to the actual number of tested genera (218 and 116 for the diarrhea and obesity datasets, respectively). For each replicate we sampled the number of times the genus was sampled across datasets. Note that to simplify this analysis we ignored the directionality of the significance (e.g., whether it was higher in case or control samples). We also excluded genera never found to be significant. We computed the mean of these 1000 distributions to generate an empirical distribution of the expected mean number of studies where a genus would be called as significant, given random sampling. We determined where the observed mean values lay on each corresponding distribution to calculate statistical significance.

**Discriminatory analysis**. We calculated the discriminatory value of each ASV (i.e., the extent to which the ASV can be used to distinguish the sample groups) based on the area under the curve (AUC) of the receiver operator curve (ROC) for that ASV. This was performed independently for both non-rarefied relative and CLR abundances. For each ASV in a dataset the abundance of that ASV along with metadata groupings was used as input into the prediction function in the ROCR R package[81]. Multiple different optimal abundance cut-offs were then used to classify samples based on the input ASVs abundance. Classifications were then compared to the true sample groupings to generate ROCs for each ASV within the 38 tested datasets. For each tool the mean AUC of all ASVs identified as being differentially abundant in each dataset was computed, based on both relative and CLR abundances separately. We then calculated the precision, recall and F1 scores of each tool for the tested datasets when AUC cut-offs of 0.7 or 0.9 were used. In each case the 'true positives' were treated as features that were above the specified AUC threshold.

**Reporting summary**. Further information on research design is available in the Nature Research Reporting Summary linked to this article.

## Data availability

The accessions and/or locations of the raw data for each tested dataset are listed in Supplementary Data 1. SILVA database v138 is available at: https://www.arb-silva.de/documentation/release-138/. Source data are provided with this paper. It is also available at https://github.com/nearinj/Comparison_of_DA_microbiome_methods[82]. The processed datasets and metadata files are available at https://figshare.com/articles/dataset/16S_rRNA_Microbiome_Datasets/14531724. Source data are provided with this paper.

## Code availability

All code used for processing and analyzing the data is available in this GitHub repository [https://github.com/nearinj/Comparison_of_DA_microbiome_methods][82].

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

## Acknowledgements

We would like to thank Mira Latva and Vinko Zadjelovic for their feedback on our manuscript. We would also like to thank everyone who responded to MGIL's queries on Twitter regarding which differential abundance tools to evaluate. Last, we would like to thank the authors of all DA tools and datasets used in this study for making their code and data freely available. JTN is funded by both a Nova Scotia Graduate Scholarship and a ResearchNS Scotia Scholars award. GMD was funded by a Canadian Graduate Scholarship (Doctoral) from NSERC. MGIL is funded through a National Sciences and Engineering Research Council (NSERC) Discovery Grant and the Canada Research Chairs program.

## Author contributions

J.T.N. and M.G.I.L. proposed the project; M.G.I.L. supervised the project; J.T.N., G.M.D., M.H., N.A., C.M.A.J., R.W., A.D., A.M.C. and M.G.I.L. collected, processed, and prepared datasets to be run through our D.A. tool pipeline. G.M.D. and J.M. contributed to the code to run all D.A. tools to produce a first-pass validation of the D.A. tools. J.T.N. expanded this approach by designing and writing the D.A. tool pipeline used in this manuscript, and also ran all datasets through this pipeline. D.K.D. created the Docker image for running the various D.A. tools and recreating the analyses. J.T.N. and G.M.D. jointly conducted all the statistical analyses and wrote the main draft of the manuscript. MH designed Table 1 and wrote parts of the Methods section. All authors provided critical feedback throughout manuscript preparation.

## Competing interests

The authors declare no competing interests.
