## [Peer Review File · Nature Communications]

Microbiome differential abundance methods produce disturbingly different results across 38 datasetReviewers' Comments:

Reviewer #1:

Remarks to the Author:

This report attempts to address an important, ongoing and heavily debated question in the microbiome field, namely which sequence data analysis programs provide the most reliable and robust microbial differential abundance analysis that results in microbial profiles that are most closely representative of the actual composition of the samples being tested. To tackle this problem, the authors analyzed 38 16S rRNA gene datasets, each with two sample groups for differential abundance testing. Abundance tables of OTUs or amplicon sequence variants (ASVs) were compiled, either provided by the original previously published studies or processed by the authors from raw sequences in a few published studies. For each dataset, the table was used to generate a rarefied or non-rarefied ASV abundance table, as well as a table where a prevalence cut-off filter of 10% was applied. Each dataset was analyzed by 14 different methods, using whichever abundance table was most suitable for the program. A significance alpha-value cutoff of 0.05 was used, and for methods that provided output p-values, false-discovery-rate (FDR)-corrected p-values were also used. The differential abundance tools and data input tested were: ALDEx2 (counts), ANCOM-II (counts), corncob (counts), DESeq2 (counts), edgeR (counts), LefSe (rarefied relative abundance), MaAsLin2 (counts), MaAslin2 (rarefied counts), metagenomeSeq (counts), limma voom (TMM counts), limma voom (TMMwsp counts), t-test (rarefied counts), Wilcoxon (CLR abundances), and Wilcoxon (rarefied counts). To compare each method, the authors generated a heatmap of the number of significant ASV hits found for each tool and Spearman correlations were determined. The authors compared the consistency between different tools within all datasets by pooling all significant ASVs and tallying the number of methods that identified each ASV as differentially abundant. To determine false discovery rates (FDRs), 8 datasets with the largest sample sizes were compared and the number of differentially abundant ASVs identified by each tool assessed at a 0.05 alpha value. The authors also performed a cross-study differential abundance consistency analysis using diarrhea and non-diarrhea sample groups and an obesity dataset comparison by determining which genera were significantly different at an alpha-value of 0.05. Overall, the manuscript is fairly well written and the study is relatively comprehensive in terms of datasets and methods applied to address an important question. However, there are a number of serious issues and concerns about the experimental design, lack of rigor regarding testing and supporting mathematical premise, and narrow scope of data interpretation that need to be addressed to provide more clarity and convincing support for the conclusions made.

1. Lines 31-33. If sample size and sequencing depth were determining factors, then what was the impact of chimera read removal? Wouldn't that impact the outcomes then, if as the authors suggest that sequence depth was a contributing factor in the inconsistency observed among methods? While the authors may not be able to test this using the preprocessed data, which removed suspected chimeric reads or so-called rare reads, they could do so using the raw data to determine if chimera removal impacted the sample size, composition and/or sequencing depth.
2. Lines 35-40. The authors need to more clearly define what they mean by "more reliable" in comparing these methods. Just because the tools cluster together does not mean that the microbial profiles obtained using them are more closely representative of the actual microbial composition of the samples or that the differential abundances are going to be discriminatory regarding biological outcomes.
3. Why did ALDEx2 and ANCOM cluster together most reliably? What is the mathematical basis for this congruence? If data is routinely removed from analysis in the pipeline, then it stands to reason that certain methods would result in more consistent outputs than others. Likewise, if LefSe, limma voom and edgeR produced inconsistent results and poorly controlled FDRs, what is the mathematical basis for this inconsistency? What differentiates these methods from ALDEx2 and ANCOM? In order for any method to be universally applicable for a wide variety of samples and their microbial communities, there must be clearly defined mathematical bases for differences observed in the outcomes.
4. Figures 1 and 2. Lines 364-366, 370-375, 380-384. The authors chose to compare the different differential abundance methods by determining the number of significant features/ASVs identified by

each method without or with prevalence filtering. Clearly the fact that the percentage of significant ASVs obtained by each method varied so widely across datasets strongly suggests that there is something inherently problematic with the approaches used to make this assessment. It is not surprising that the authors found several contributing factors. The authors do not provide any insight as to why the different methods gave such widely disparate outcomes in identifying significant features. What is the mathematical basis for these different outcomes? What does the finding that there are more significant ASV hits mean in terms of relevance to the microbial profiles of the samples? Are they more similar or more meaningful in terms of discriminating disease outcome? Importantly, in none of the analyses do the authors consider the key question: How closely do the resulting microbial composition profiles obtained by each of the methods resemble each other when considering discriminating outcomes within each dataset versus among multiple datasets?

5. Figure 3, lines 502-503. The authors assume that having an overlap in significant ASVs indicates that the tools provide congruent outcomes and that those methods are more reliable and consistent, despite having quite different methodological approaches. As noted by the authors, each of the methods emphasize different aspects of the sample's microbial composition. However, they do not provide any mathematical backing of these conclusions or a rationale for why this might be the case. As the authors point out (lines 513-515), there is substantial missing information that contributes to the observed clustering patterns.

6. One critical question that the authors did not address when comparing each of the methods for analyzing the datasets is: Which method provided microbial composition profiles that more closely represented the expected microbial community? Why did the authors not compare their methods using the mockrobiota community raw dataset available for researchers (PMID: 27822553)? At least with these datasets, which have defined microbial compositions for each of the sample datasets, it should be possible for the authors to determine which methods provides a microbial profile that more closely resembles the expected defined community. Then, any differences observed between sample types within a dataset (such as healthy versus diseased) should provide more reliable and robust differential abundance profiles.

7. Lines 166-169. Another critical question that the authors did not address is: What about chimeric reads? In all of the datasets used for this study, the raw sequencing reads were processed using QIIME2, which routinely removes any suspected chimeric reads from the dataset before further analyses are performed. Depending on the sample and its complexity, this step could significantly impact the number of reads for any given sample, and thereby, could impact the outcomes after processing through each of the differential abundance methods. What were the percentage of chimeric reads removed for each of the samples in these studies? If there was more than 5-10% of the reads removed, it is difficult to believe that this would not impact the sequence depth, complexity, and diversity of the samples, and therefore the outcomes.

8. Lines 337-339, Table 2, Figure 5. What does this statement mean? Were these "two additional datasets" not listed as part of the original 38? What are the literature citations for these studies? Were these datasets part of published studies? The authors need to clarify which studies were used for this cross-study analysis, i.e., specify which studies were included in this analysis and provide the corresponding literature citations.

9. Lines 343-344. The authors should specify exactly which obesity datasets were used for this analysis and provide the corresponding literature citations.

10. In terms of choice of datasets, there have been in recent years at least three seminal, large-scale studies regarding the impact of gut microbiomes on cancer interventions (PMID:29097493; PMID:29097494; PMID:29302014) for which the significant ASVs were found to be completely different between the responsive versus nonresponsive cohorts. There is no mention of these studies in the discussion or the impact of the current study to address issues raised by those other studies.

11. Did any of the methods account for the complexity of the community of the sample in determining significant ASVs, as opposed to just considering prevalence or low-counts for cut-off thresholds?

Reviewer #2:

Remarks to the Author:

The authors present a comparison between most popular differential abundance methods for the microbiome based on a range of datasets. The great divergence of methods and lack of reproducibility of the results is correctly pointed out as a major problem, making this a timely and relevant work. The reliance on real data rather than simulation lends great credibility to the results. The manuscript is well written and demonstrates good literature review. The availability of data and code for reproducibility is exemplary. However, the description of the methods used and the evaluation methodology still needs improvement.

Major comments

* The main criterion for method evaluation is consistency of the results over the different datasets. Yet as the authors acknowledge, this benefits conservative methods: picking up only the strongest signal is the safest way to report reproducible results. Decreasing the significance level for any method will increase consistency. I believe an additional criterion is needed that also rewards more powerful methods that stick their neck out and report taxa with lower effect sizes too. I suggest the authors look at either consistency of taxon rankings, or take the twenty or so most significant genera and compare them across the experiments. This latter way reflects the way many scientists proceed when confronted with a long list of hits which they cannot all investigate further. Another criterion is whether the FDR truly lies around the significance level of 0.05. In relation to this, the authors should specify how genera not present in all studies are treated (I336), and how they affect the results.

* In relation to this, I wonder whether the Kolmogorov-Smirnov test is the best choice for testing consistency. It uses the supremum of the difference between expected and observed distribution function to calculate the p-value (last two columns of table 2). My first question is why to focus on the supremum. The second is how this test statistic relates to the fold change between mean and observed number of datasets? I think the latter is indeed a useful measure, but then the statistical test should apply to this quantity directly. The resampling procedure outlined by the authors could be used to establish its null distribution. A section on this test in the supplementary material would lessen confusion.

* I650 "While it might be argued that differences in tool outputs are expected given that they different hypotheses, we believe this perspective ignores how these tools are used in practice."

I agree that most scientists ignore subtle differences in hypotheses tested by the different methods, but in the context of this manuscript it would be useful to elaborate on them, reiterate them when discussing results. This may help explain the difference in results, e.g. why CoDa methods like ANCOM and ALDEX2 are more conservative.

* I531: In this complete null scenario, every discovery is a false discovery, and hence the false discovery proportion (FDP) in a single Monte-Carlo instance is either 0 (no discoveries) or 1 (one or more discoveries). The estimated FDR is then equal to the FWER, and is estimated as the average FDP over the Monte-Carlo instances. This number should hover around 0.05. I am not sure if this is the way the authors proceeded, but some more explanation may lessen confusion. Also, I would recommend doing at least 100 Monte-Carlo instances to obtain reliable results.

* Personally, I would find it impractical to combine several differential abundance methods on a single dataset. It implies more work, the hypotheses tested may differ, and making choices on how exactly to pool the results is arbitrary. But that is a matter of taste, I agree that it is hard to make positive recommendations.

* As a last major comment, the description of the different differential abundance methods in general needs improvement. In addition, some methods should be revised as they depart too far from the default specified by the authors. Detailed comments can be found below:

- I80: "... are both tools that assume normalized read counts follow a negative binomial distribution".

I do not agree: the raw read counts are assumed to follow the negative binomial distribution. The parameters of this distribution may depend on an estimate of the sequencing depth or other regressors, but the data are not normalised by them.

- I82: "The null hypothesis is that the same parameters for the binomial solution explain the distribution of taxa across all sample groupings."

I find this a little inaccurate. It is true that it is assumed that the dispersion parameter is constant over groups, but this is not part of the null hypothesis tested. The null hypothesis usually refers to a single parameter or a group of dummy parameters being equal (often equal to zero), not necessarily to all parameters.

- I221: "This function first converted the into relative abundances and then fit each taxon abundance to a beta-binomial model"

The beta-binomial model is a count model, so here as well I believe the model is fitted on the raw counts.

- I266: I find this a strange way of running limma-voom. In the documentation it asks for a count matrix, but it seems that the authors prenormalised the data. If this is true they should justify why they depart from the documented limma pipeline, but I would rather see a count matrix being fed to it. Moreover, can the authors motivate their choice of normalization method, i.e. why was limma-voom combined with two normalization methods and the other methods not?

- I297: For t-test and Wilcoxon rank sum test, it would be good to include tests on relative abundance, without rarefying. I expect this to be more powerful, and also often happens in this way. As a minor remark, it is pointless to apply total sum scaling to rarefied data as mentioned for the t-test, as all sample sums are equal after rarefying (it should not affect the results though).

- I483: "Furthermore, the two Wilcoxon test approaches had different consistency profiles despite the same hypothesis test."

Since one test is on rarefied abundances and the other on a log-ratio, these can hardly be called the same hypothesis. Changes in other taxa abundances may affect the log-ratio, and thus invalidate the null hypothesis, whereas the null hypothesis of equal abundances remains valid. This can also be seen from Figures C and D, where t-test and Wilcoxon rank sum test after rarefying are really close as they test very similar hypotheses, as opposed to Wilcoxon CLR. I suspect a t-test CLR would also cluster close to the latter.

Minor comments

* I554: "we also observed that in several replicates on the unfiltered datasets, the Wilcoxon (CLR) approach identified almost all features as statistically significant"

Is this a consequence of a variable geometric mean, which is the denominator in the CLR transformation?

* I762 "Indeed, tests based on rarefied data in our analyses did not substantially worse than other methods on average."

It is hard to compare with the given results, one would need to compare the same method with and

without rarefying, which was not done for most methods.

* I517 "Interestingly, corncob, which is a methodologically distinct approach, also clusters relatively close to these two methods on the first PC."

At parameter values encountered in microbiome data, the beta-binomial and negative binomial distributions are identical in terms of density. Hence it is not so surprising that corncob and edgeR are close.

* p17 Figures A and B: this graph is interesting, but good to mention it strongly depends on which methods were included, and how many times (e.g. limma and Maaslin in duplicate).

* I632 each genera -> each genus

* Supplementary Figures (and Supplementary Table 1) are sometimes hard to read. Perhaps a tool like knitr or rmarkdown can help to embed figures in a text document.

Reviewer #3:

Remarks to the Author:

Nearing et al. conducted an in-depth and a comprehensive evaluation of common tools used to identify differentially abundant taxa in 16S microbiome studies. They performed their analysis on large dataset of 38 two-group 16S rRNA gene sequencing datasets. The authors looked at the concordance of the methods on these datasets with and without filtration of rare taxa. They additionally looked at the observed FDR for each tool and lastly looked at how consistent biological interpretations would be by looking across diarrheal datasets. Overall, the manuscript is well-written, and the analyses performed are appropriate.

The manuscript addresses an issue that is often overlooked by so many in the field of microbiome research. While the conclusion about inconsistencies across tools is disturbing, it is not completely surprising. I have a few concerns with the manuscript primarily with how the data is presented and how the conclusions are discussed or outlined.

- I think the manuscript would benefit if the authors showed analysis of a selection of simulations. In my opinion, this would make the conclusions more understandable to the readers.
- While it is not easy to make a ruling on which tools would be best as acknowledged by the authors, I think there should be some recommendation for gold standards for the different applications. I would suggest adding a summary scheme or Box in which "advantages vs disadvantages" or "recommendations for use for different applications" of the tools used are summarized. This would make this manuscript more accessible to clinicians/microbiologists who are using these tools. As it stands now, I find the manuscript confusing/complicated to understand for the non-experts who are using DA tools to identify taxa relevant to diseases/phenotypes.

REVIEWER COMMENTS

Reviewer #1 (Expertise: Analysis of complex microbiome data):

This report attempts to address an important, ongoing and heavily debated question in the microbiome field, namely which sequence data analysis programs provide the most reliable and robust microbial differential abundance analysis that results in microbial profiles that are most closely representative of the actual composition of the samples being tested. To tackle this problem, the authors analyzed 38 16S rRNA gene datasets, each with two sample groups for differential abundance testing. Abundance tables of OTUs or amplicon sequence variants (ASVs) were compiled, either provided by the original previously published studies or processed by the authors from raw sequences in a few published studies. For each dataset, the table was used to generate a rarefied or non-rarefied ASV abundance table, as well as a table where a prevalence cut-off filter of 10% was applied. Each dataset was analyzed by 14 different methods, using whichever abundance table was most suitable for the program.

A significance alpha-value cutoff of 0.05 was used, and for methods that provided output p-values, false-discovery-rate (FDR)-corrected p-values were also used. The differential abundance tools and data input tested were: ALDEx2 (counts), ANCOM-II (counts), corncob (counts), DESeq2 (counts), edgeR (counts), LefSe (rarefied relative abundance), MaAsLin2 (counts), MaAslin2(rarefied counts), metagenomeSeq (counts), limma voom (TMM counts), limma voom (TMMwsp counts), t-test (rarefied counts), Wilcoxon (CLR abundances), and Wilcoxon (rarefied counts). To compare each method, the authors generated a heatmap of the number of significant ASV hits found for each tool and Spearman correlations were determined. The authors compared the consistency between different tools within all datasets by pooling all significant ASVs and tallying the number of methods that identified each ASV as differentially abundant. To determine false discovery rates (FDRs), 8 datasets with the largest sample sizes were compared and the number of differentially abundant ASVs identified by each tool assessed at a 0.05 alpha value. The authors also performed a cross-study differential abundance consistency analysis using diarrhea and non-diarrhea sample groups and an obesity dataset comparison by determining which genera were significantly different at an alpha-value of 0.05. Overall, the manuscript is fairly well written and the study is relatively comprehensive in terms of datasets and methods applied to address an important question. However, there are a number of serious issues and concerns about the experimental design, lack of rigor regarding testing and supporting mathematical premise, and narrow scope of data interpretation that need to be addressed to provide more clarity and convincing support for the conclusions made.

We thank the reviewer for their detailed review of our work. We appreciate several of their specific suggestions, but we believe that there may be a misunderstanding regarding the scope and focus of our manuscript.

In particular, the first sentence of this reviewer's report was the following:

"This report attempts to address an important, ongoing and heavily debated question in the microbiome field, namely which sequence data analysis programs provide the most reliable and robust microbial differential abundance analysis that **results in microbial profiles that are most closely representative of the actual composition of the samples being tested.**"

The tools we applied are solely intended to be used for differential abundance testing and the same input datasets were used for each tool. This means that our goal was not to assess processing pipelines to see how accurately they produced expected profiles, as is a typical approach when comparing pipelines such as QIIME 2 and mothur with mock communities for instance. Our goals here were qualitatively different and we believe it would be tangential to our manuscript's purpose to compare such pipelines or what factors might impact the final sequencing depth.

1. Lines 31-33. If sample size and sequencing depth were determining factors, then what was the impact of chimera read removal? Wouldn't that impact the outcomes then, if as the authors suggest that sequence depth was a contributing factor in the inconsistency observed among methods? While the authors may not be able to test this using the preprocessed data, which removed suspected chimeric reads or so-called rare reads, they could do so using the raw data to determine if chimera removal impacted the sample size, composition and/or sequencing depth.

As described above, the same input datasets (whether prevalence filtered or not) were used to test each tool, so we do not believe that a deep investigation into chimeras is warranted for our manuscript's purposes.

Nonetheless, we ran chimera checking with the UCHIME algorithms (implemented in VSEARCH) to estimate what percentage of ASVs/OTUs in each dataset were chimeric and we now mention this in the text. Importantly, no ASVs/OTUs were identified as chimeric (based on comparisons of the final feature sequences) based on de novo chimera checking approaches. However, based on reference-based chimera picking with the UCHIME2 algorithm, there were in some cases high percentages of chimeric features (e.g., >30% in one case). In several cases the percentage of significant features output by each tool was moderately correlated with the percentage of chimeric features. We have included a new supplementary figure displaying these results (**Supp Fig 1**).

In addition, we believe the reviewer picked up on the interesting notion that the percent of rare ASVs in each dataset, which could be associated with artifacts such as chimeras, might help explain the variation in the number of significant ASVs. Accordingly, we added to our heatmaps (**Fig. 1**) the mean percent of ASV abundances that were removed from each sample following our 10% prevalence cut-off. In several cases this dataset characteristic was also significantly associated with the percentage of significant ASVs identified by each tool.

2. Lines 35-40. The authors need to more clearly define what they mean by "more reliable" in comparing these methods. Just because the tools cluster together does not mean that the microbial profiles obtained using them are more closely representative of the actual microbial composition of the samples or that the differential abundances are going to be discriminatory regarding biological outcomes.

In this particular case we were referring to the recommendations made by a separate study that claimed that limma voom, corncob, and DESeq2 were more reliable. We have changed this wording to be "more accurate", which was based on numerous criteria in a separate study that we cite (Calgaro et al. 2020).

However, we do agree that in our work that just because the results of certain tools are more consistent with others does not mean that they are necessarily more reliable. We now clarify this at the beginning of the results section presenting those consistency results:

"These analyses provided insight into how similar the interpretations would be depending on which DA method was applied. We hypothesize that tools that produce significant ASVs that highly intersect with the output of other DA tools are the most accurate approaches. However, this is not necessarily the case: tools that produce similar sets of significant ASVs could simply be picking up on the same noise and producing similar sets of false positives. This caveat should be noted while interpreting these results."

3. Why did ALDEx2 and ANCOM cluster together most reliably? What is the mathematical basis for this congruence? If data is routinely removed from analysis in the pipeline, then it stands to reason that certain methods would result in more consistent outputs than others. Likewise, if LEfSe, limma voom and edgeR produced inconsistent results and poorly controlled FDRs, what is the mathematical basis for this inconsistency? What differentiates these methods from ALDEx2 and ANCOM? In order for any method to be universally applicable for a wide variety of samples and their microbial communities, there must be clearly defined mathematical bases for differences observed in the outcomes.

ALDEx2 and ANCOM-II are both compositional data analysis tools that examine abundance values using ratios rather than relative abundances. Although they use different dominators in their ratio approaches (CLR vs ALR as we discuss in the introduction), they can be considered to be within the same overall method class. For this reason we believe that they tend to cluster together. We believe that the question of what are the statistical and/or mathematical reasons for the inconsistent results output by certain tools like LEfSe, limma voom, and edgeR, is crucial for further development of improved differential methods, but that is beyond the scope of our manuscript. Rather than comparing the algorithm details, our work has been from the user's perspective. In other words, we investigated whether these tools should be used interchangeably. Clearly they should not be based on our results and we discuss certain high-level reasons why the tools differ, such as the advantage of compositional data analysis methods in certain cases and the downside of tools like edgeR and limma voom that may fail when datasets have high sparsity. We believe this practical

level of understanding how the tools perform is most relevant to microbiome researchers and that further contrasting of mathematical differences between the tools would make the paper much more difficult to follow and thus less impactful.

4. Figures 1 and 2. Lines 364-366, 370-375, 380-384. The authors chose to compare the different differential abundance methods by determining the number of significant features/ASVs identified by each method without or with prevalence filtering. Clearly the fact that the percentage of significant ASVs obtained by each method varied so widely across datasets strongly suggests that there is something inherently problematic with the approaches used to make this assessment. It is not surprising that the authors found several contributing factors. The authors do not provide any insight as to why the different methods gave such widely disparate outcomes in identifying significant features. What is the mathematical basis for these different outcomes? What does the finding that there are more significant ASV hits mean in terms of relevance to the microbial profiles of the samples? Are they more similar or more meaningful in terms of discriminating disease outcome? Importantly, in none of the analyses do the authors consider the key question: How closely do the resulting microbial composition profiles obtained by each of the methods resemble each other when considering discriminating outcomes within each dataset versus among multiple datasets?

We appreciate the reviewer's comments and questions. There are a few different points to respond to here.

First, regarding mathematical bases for differences you can see our response to the related point #3 above.

Regarding how having more significant ASV hits could be relevant to the specific microbial samples - the variation in the number of significant ASV hits is related to the characteristics of each microbiome dataset, as indicated on Figures 1 and 2 by metadata such as the dataset sparsity. It is also related to the overall signal differentiating the sample groups, as indicated by the PERMANOVA coefficients.

Regarding whether the ASVs are more relevant for discriminating disease outcome (and more generally the tested sample groups from any environment) - we now include an additional set of analyses to get at this question. Specifically, we have evaluated how useful each ASV is on its own for discriminating between the sample groups. Our new set of analyses that we now describe in the results (and with added supplementary figures (**Sup Figs. 3-4**) is based on comparing how useful the sets of significant ASVs identified by each tool are based on this approach.

Last, regarding the reviewer's last question here, we believe this is related to the confusion regarding what these tools are producing: the microbial composition profiles are the input to the tools and were the same across all tools (except for rarefied vs non-rarefied and filtered vs unfiltered datasets).

5. Figure 3, lines 502-503. The authors assume that having an overlap in significant ASVs indicates that the tools provide congruent outcomes and that those methods are more reliable and consistent, despite having quite different methodological approaches. As noted by the authors, each of the methods emphasize different aspects of the sample's microbial composition. However, they do not provide any mathematical backing of these conclusions or a rationale for why this might be the case. As the authors point out (lines 513-515), there is substantial missing information that contributes to the observed clustering patterns.

Similar to our response to the reviewer's point #2 we think this is an important point and have clarified in the text that just because tools overlap with others does not mean that they are more reliable:

“These analyses provide insight into how similar the results of different tools are expected to be, which could be due to methodological similarities between them. However, this does not provide clear evidence for which tools are the most accurate.”

6. One critical question that the authors did not address when comparing each of the methods for analyzing the datasets is: Which method provided microbial composition profiles that more closely represented the expected microbial community? Why did the authors not compare their methods using the mockrobiota community raw dataset available for researchers (PMID: 27822553)? At least with these datasets, which have defined microbial compositions for each of the sample datasets, it should be possible for the authors to determine which methods provides a microbial profile that more closely resembles the expected defined community. Then, any differences observed between sample types within a dataset (such as healthy versus diseased) should provide more reliable and robust differential abundance profiles.

As described above, our manuscript focuses solely on comparing different differential abundance methods and not in comparing workflows for producing microbial composition profiles. Because of this we disagree that the analyses the reviewer describes would be relevant in this case. However, we certainly think this point touches on important questions related to other aspects of microbiome data analysis, but that are outside the scope of our paper.

7. Lines 166-169. Another critical question that the authors did not address is: What about chimeric reads? In all of the datasets used for this study, the raw sequencing reads were processed using QIIME2, which routinely removes any suspected chimeric reads from the dataset before further analyses are performed. Depending on the sample and its complexity, this step could significantly impact the number of reads for any given sample, and thereby, could impact the outcomes after processing through each of the differential abundance methods. What were the percentage of chimeric reads removed for each of the samples in these studies? If there was more than 5-10% of the reads removed, it is difficult

to belief that this would not impact the sequence depth, complexity, and diversity of the samples, and therefore the outcomes.

Please see our response to the reviewer's point #1 regarding the prevalence of chimeric ASVs where we discuss this issue.

8. Lines 337-339, Table 2, Figure 5. What does this statement mean? Were these “two additional datasets” not listed as part of the original 38? What are the literature citations for these studies? Were these datasets part of published studies? The authors need to clarify which studies were used for this cross-study analysis, i.e., specify which studies were included in this analysis and provide the corresponding literature citations.

Thanks for catching this. Those two datasets were not part of the origin 38, but were part of published studies. We now clarify all of the questions you raised in that section of the methods:

“For this analysis we acquired two additional pre-processed datasets that were not used for other analyses, which are the GEMS1 (Pop et al., 2014) and the dia_schneider (Schneider et al., 2017), datasets (Supp. Table 1). The processed data for these datasets was acquired from the MicrobiomeDB (Oliveira et al., 2018) and the microbiomeHD (Duvall et al., 2017) databases, respectively. These datasets were combined with three of the datasets used elsewhere in this manuscript (Human – C. diff [1 and 2] and Human – Inf.), to bring the number of diarrhea-related datasets to five. These three pre-existing datasets all related to enteric infections that had all been previously shown to show a distinct signal of microbial differences driven by diarrhea in patient samples (Duvall et al., 2017).”

9. Lines 343-344. The authors should specify exactly which obesity datasets were used for this analysis and provide the corresponding literature citations.

We now provide these details in that section of the methods:

“For the obesity cross-study analysis we leveraged four datasets that were part of core 38 datasets: Human – OB (1-4) (Goodrich et al., 2014; Ross et al., 2015; Turnbaugh et al., 2009; Zhu et al., 2013). We also included an additional obesity dataset, ob_zupancic (Zupancic et al., 2012), that we acquired from the microbiomeHD database.”

0. In terms of choice of datasets, there have been in recent years at least three seminal, large-scale studies regarding the impact of gut microbiomes on cancer interventions (PMID:29097493; PMID:29097494; PMID:29302014) for which the significant ASVs were found to be completely different between the responsive versus nonresponsive cohorts. There is no mention of these studies in the discussion or the impact of the current study to address issues raised by those other studies.

Thanks for pointing us to these studies. We now include this sentence near the beginning of our discussion:

“These results might partially account for the common observation that significant microbial features reported in one dataset only marginally overlap with significant hits in similar datasets, such as recently found across three high-profile cancer intervention studies (Gopalakrishnan et al., 2018; Matson et al., 2018; Routy et al., 2018). However it should be noted that this could also be due to numerous other biases that are present in microbiome studies (Nearing et al., 2021).”

11. Did any of the methods account for the complexity of the community of the sample in determining significant ASVs, as opposed to just considering prevalence or low-counts for cut-off thresholds?

Several methods to account for community complexity, which we now highlight in the discussion:

...“the variance in the number of features identified could also be attributed to pre-processing steps various tools use to remove potential ASVs for testing. This includes corncob that does not report significance values for ASVs only found in one group or ANCOM-II’s ability to remove structural zeros”

Reviewer #2 (Expertise: Statistical methods for the analysis of 16S data, microbiome studies):

The authors present a comparison between most popular differential abundance methods for the microbiome based on a range of datasets. The great divergence of methods and lack of reproducibility of the results is correctly pointed out as a major problem, making this a timely and relevant work. The reliance on real data rather than simulation lends great credibility to the results. The manuscript is well written and demonstrates good literature review. The availability of data and code for reproducibility is exemplary. However, the description of the methods used and the evaluation methodology still needs improvement.

Major comments

* The main criterion for method evaluation is consistency of the results over the different datasets. Yet as the authors acknowledge, this benefits conservative methods: picking up only the strongest signal is the safest way to report reproducible results. Decreasing the significance level for any method will increase consistency. I believe an additional criterion is needed that also rewards more powerful methods that stick their neck out and report taxa with lower effect sizes too. I suggest the authors look at either consistency of taxon rankings, or take the twenty or so most significant genera and compare them across the

experiments. This latter way reflects the way many scientists proceed when confronted with a long list of hits which they cannot all investigate further. Another criterion is whether the FDR truly lies around the significance level of 0.05. In relation to this, the authors should specify how genes not present in all studies are treated (I336), and how they affect the results.

Thanks for the suggestion - we agree that in principle the top ASVs of all tools could have been similar and that some tools might simply stick their necks out more as you said. To get at this question we looked at the overlap in the top 20 ASVs per tool per dataset and describe the results in this new paragraph of the results:

“A major caveat of the above analysis is that each DA tool produced substantially different numbers of ASVs in total. Accordingly, in principle all the tools could be identifying the same top ASVs and simply taking varying degrees of risk when identifying less clearly differential ASVs. To investigate this possibility, we identified the overlap between the 20 top-ranked ASVs for each tool per dataset (Supp. Fig 5), which included non-significant (but relatively highly ranked) ASVs in some cases. The distribution of these ASVs in the 20 top-ranked hits for all tools was similar to that of all significant ASVs described above. For example, in the filtered data we found that, t-test (rare) (mean: 7.7), edgeR, (mean: 7.0), corncob (mean: 6.8), and DESeq2 (mean: 5.4) had the highest number of ASVs only identified by that particular tool as being in the top 20. Only a small number of ASVs were amongst the top 20 ranked of all tools in both the filtered (mean: 2.0) and unfiltered (mean: 1.0) datasets (Supp. Fig 5).”

* In relation to this, I wonder whether the Kolmogorov-Smirnov test is the best choice for testing consistency. It uses the supremum of the difference between expected and observed distribution function to calculate the p-value (last two columns of table 2). My first question is why to focus on the supremum. The second is how this test statistic relates to the fold change between mean and observed number of datasets? I think the latter is indeed a useful measure, but then the statistical test should apply to this quantity directly. The resampling procedure outlined by the authors could be used to establish its null distribution. A section on this test in the supplementary material would lessen confusion.

We agree that comparing the supremum of the expected and observed distributions could be non-optimal for our purposes. We implemented the test you outlined (i.e., determining significance of the observed mean based on a distribution of expected means) and the results are qualitatively the same (except that ALDEx2 is now also significant across the obesity datasets). We now present the results of this analysis in the results as we believe it will be easier for readers to follow and is not based on the supremum of the distributions alone.

* I650 "While it might be argued that differences in tool outputs are expected given that they different hypotheses, we believe this perspective ignores how these tools are used in practice."

I agree that most scientists ignore subtle differences in hypotheses tested by the different methods, but in the context of this manuscript it would be useful to elaborate on them, reiterate them when discussing results. This may help explain the difference in results, e.g. why CoDa methods like ANCOM and ALDEX2 are more conservative.

We now describe the likely reasons why ANCOM-II and ALDEX2 appear to be more conservative in the discussion section.

* I531: In this complete null scenario, every discovery is a false discovery, and hence the false discovery proportion (FDP) in a single Monte-Carlo instance is either 0 (no discoveries) or 1 (one or more discoveries). The estimated FDR is then equal to the FWER, and is estimated as the average FDP over the Monte-Carlo instances. This number should hover around 0.05. I am not sure if this is the way the authors proceeded, but some more explanation may lessen confusion. Also, I would recommend doing at least 100 Monte-Carlo instances to obtain reliable results.

We agree that this would be one way to evaluate the tools, but our analysis instead focused on the mean percentage of ASVs that were called as significant. We have added text at the beginning of this section to clarify that that is what we are focused on (in the main text at least). In this context and ASVs called as significant is indeed a problem, but what's striking is that in some cases tools virtually always called some ASVs as significant (even if just a low percentage). In contrast, other tools almost never called significant features, but when they did they called a substantial proportion as significant. We believe both of these trends can be gleaned by considering both our heatmap of mean % significant ASVs in addition to the breakdown shown per Monte-Carlo instance in the supplementary materials.

In addition, we have also run additional Monte-Carlo instances for long-running tools to ensure that those results are reliable.

* Personally, I would find it impractical to combine several differential abundance methods on a single dataset. It implies more work, the hypotheses tested may differ, and making choices on how exactly to pool the results is arbitrary. But that is a matter of taste, I agree that it is hard to make positive recommendations.

We completely agree that a consensus approach complicates analyses and the interpretation. However, as we outline in the text, we believe that moving towards consensus-based approaches is needed until the field settles on a single DA approach to use.

In addition, we agree that the hypotheses being tested can differ between the tools, but importantly they are usually treated interchangeably in the microbiome literature. For this reason we think that providing differential abundance results based on a consensus of different tools would be the most conservative way to compare across studies.

The exception is if researchers know specifically what studies they want to compare with then using the same DA approaches may be the best way forward. We have added this important point to our discussion:

“If researchers are most interested in determining if signals from another study are reproducible, then they should ensure that they use the same DA approach.”

* As a last major comment, the description of the different differential abundance methods in general needs improvement. In addition, some methods should be revised as they depart too far from the default specified by the authors. Detailed comments can be found below:

- I80: "... are both tools that assume normalized read counts follow a negative binomial distribution".

I do not agree: the raw read counts are assumed to follow the negative binomial distribution. The parameters of this distribution may depend on an estimate of the sequencing depth or other regressors, but the data are not normalised by them.

Thanks for catching this. We now specify that the raw read counts, not the normalized reads, are assumed to follow a negative binomial distribution.

- I82: "The null hypothesis is that the same parameters for the binomial solution explain the distribution of taxa across all sample groupings."

I find this a little inaccurate. It is true that it is assumed that the dispersion parameter is constant over groups, but this is not part of the null hypothesis tested. The null hypothesis usually refers to a single parameter or a group of dummy parameters being equal (often equal to zero), not necessarily to all parameters.

We have now changed this sentence to read:

“The null hypothesis is that the same setting for certain parameters of the negative binomial solution explain the distribution of taxa across all sample groupings.”

- I221: "This function first converted the into relative abundances and then fit each taxon abundance to a beta-binomial model"

The beta-binomial model is a count model, so here as well I believe the model is fitted on the raw counts.

This has been corrected.

- I266: I find this a strange way of running limma-voom. In the documentation it asks for a count matrix, but it seems that the authors prenormalized the data. If this is true they should justify why they depart from the documented limma pipeline, but I would rather see a count matrix being fed to it. Moreover, can the authors motivate their choice of normalization method, i.e. why was limma-voom combined with two normalization methods and the other methods not?

We decided to run limma-voom as it was outlined in the limma bioconductor user guide found here:

<https://www.bioconductor.org/packages/release/bioc/vignettes/limma/inst/doc/usersguide.pdf>.

Furthermore, the original paper outlining the voom method (<https://doi.org/10.1186/gb-2014-15-2-r29>) also used TMM normalization in their analysis. As such to stay consistent with the author recommendations we choose to normalize the data using TMM normalization. We also choose to test out the use of TMMwsp normalization as well due to it being highlighted as an alternative normalization method for highly sparse data as outlined within the edgeR package. We choose to test both methods as limma-voom using TMM normalization was previously shown to perform preferentially in DA testing (Calgaro et al., 2020). In our hands we did not find this to be the case, however, we noted that our datasets were of higher sparsity than those tested by Calgaro et al., and as such we decided to also test a normalization method made to specifically address issues with sparsity.

- I297: For t-test and Wilcoxon rank sum test, it would be good to include tests on relative abundance, without rarefying. I expect this to be more powerful, and also often happens in this way. As a minor remark, it is pointless to apply total sum scaling to rarefied data as mentioned for the t-test, as all sample sums are equal after rarefying (it should not affect the results though).

Thanks for this suggestion, but we chose not to include the Wilcoxon and t-test approaches based on non-rarefied relative abundance by design. This was because it has been previously shown that applying such tools without normalizing the data in some way can lead to more false inferences, but otherwise similar results to first rarefying data (e.g., [10.1186/s40168-017-0237-y](https://doi.org/10.1186/s40168-017-0237-y)). Given this past work, we do not believe that an additional comparison of rarefied and raw relative abundances based on these tools need be included in our work.

- I483: "Furthermore, the two Wilcoxon test approaches had different consistency profiles despite the same hypothesis test."

Since one test is on rarefied abundances and the other on a log-ratio, these can hardly be called the same hypothesis. Changes in other taxa abundances may affect the log-ratio,

and thus invalidate the null hypothesis, whereas the null hypothesis of equal abundances remains valid. This can also be seen from Figures C and D, where t-test and Wilcoxon rank sum test after rarefying are really close as they test very similar hypotheses, as opposed to Wilcoxon CLR. I suspect a t-test CLR would also cluster close to the latter.

Thanks for catching this - we agree entirely that the hypotheses being tested are quite different. We have now changed the text to reflect this:

“Furthermore, the two Wilcoxon test approaches had highly different consistency profiles, which highlights the impact that CLR-transforming has on downstream results.”

Minor comments

* I554: "we also observed that in several replicates on the unfiltered datasets, the Wilcoxon (CLR) approach identified almost all features as statistically significant"

Is this a consequence of a variable geometric mean, which is the denominator in the CLR transformation?

Yes, we believe that is the most likely driving factor for these observations. We have now added several statements following up on our report of differential read depth being associated with outlier samples:

“These differences were associated with similar differences in the geometric mean abundances per-sample (i.e., the denominator of the CLR transformation) between the test groups. Specifically, per dataset, these outliers commonly displayed the most extreme difference in geometric mean on average between the test groups and were otherwise amongst the top ten most extreme replicates.”

* I762 "Indeed, tests based on rarefied data in our analyses did not substantially worse than other methods on average."

It is hard to compare with the given results, one would need to compare the same method with and without rarefying, which was not done for most methods.

Thanks for pointing this out. We agree that in general to ask how rarefaction affects each method we would need to rarefy the data prior to running all DA methods. However, our intention was actually to focus on the reality that certain tools expect rarefied input data (or

are usually used that way anyway) and for others the tool authors recommend that the input data not be rarefied. We have now clarified that section of the text to reflect this message, which is a much narrower statement than what we had initially stated.

* l517 "Interestingly, corncob, which is a methodologically distinct approach, also clusters relatively close to these two methods on the first PC."

At parameter values encountered in microbiome data, the beta-binomial and negative binomial distributions are identical in terms of density. Hence it is not so surprising that corncob and edgeR are close.

Thanks for pointing this out - we now refer to this possibility:

"This may reflect that the distributions that these two methods rely upon become similar when considering the parameter values often associated with microbiome data"

* p17 Figures A and B: this graph is interesting, but good to mention it strongly depends on which methods were included, and how many times (e.g. limma and Maaslin in duplicate).

Thanks for pointing this out, we now include this caveat in the Figure 3 legend:

"Note that when interpreting these results that they are dependent on which methods were included, and whether they are represented multiple times. For instance, two different workflows for running MaAslin2 are included, which produced similar outputs."

* l632 each genera -> each genus

Changed.

* Supplementary Figures (and Supplementary Table 1) are sometimes hard to read. Perhaps a tool like knitr or rmarkdown can help to embed figures in a text document.

We have greatly cleaned up and resized Supplementary Table 1 and it is now much easier to read.

We also added a new Supplementary Table 2, which provides a brief description of each dataset, which readers may find useful.

Reviewer #3 (Expertise: Studying the gut microbiome):

Nearing et al. conducted an in-depth and a comprehensive evaluation of common tools used to identify differentially abundant taxa in 16S microbiome studies. They performed their analysis on large dataset of 38 two-group 16S rRNA gene sequencing datasets. The authors looked at the concordance of the methods on these datasets with and without filtration of rare taxa. They additionally looked at the observed FDR for each tool and lastly looked at how consistent biological interpretations would be by looking across diarrheal datasets. Overall, the manuscript is well-written, and the analyses performed are appropriate.

The manuscript addresses an issue that is often overlooked by so many in the field of microbiome research. While the conclusion about inconsistencies across tools is disturbing, it is not completely surprising. I have a few concerns with the manuscript primarily with how the data is presented and how the conclusions are discussed or outlined.

- I think the manuscript would benefit if the authors showed analysis of a selection of simulations. In my opinion, this would make the conclusions more understandable to the readers.

We deliberately avoided basing our analyses on simulations as several papers have already been published that focus specifically on validating DA tools with simulations (e.g., [10.1093/bib/bbx104](https://doi.org/10.1093/bib/bbx104) and [10.1186/s40168-016-0208-8](https://doi.org/10.1186/s40168-016-0208-8)). This work has been valuable but suffers from the limitation that the parameters used to simulate microbial profiles may not reflect true real-world samples and that tools designed around these parameters will perform better in those simulations. For this reason, we felt additional validations on actual datasets would be a useful contribution to the field. We now discuss this topic in more detail in a new paragraph of our discussion.

- While it is not easy to make a ruling on which tools would be best as acknowledged by the authors, I think there should be some recommendation for gold standards for the different applications. I would suggest adding a summary scheme or Box in which “advantages vs disadvantages” or “recommendations for use for different applications” of the tools used are summarized. This would make this manuscript more accessible to clinicians/microbiologists who are using these tools. As it stands now, I find the manuscript confusing/complicated to understand for the non-experts who are using DA tools to identify taxa relevant to diseases/phenotypes.

We appreciate that many readers will be looking for clear recommendations to follow. For this reason, we explicitly raised this point in our discussion and highlighted that certain tools are likely more specific (i.e., ALDEX2 and ANCOM-II), while other tools may be more useful for users in need of higher sensitivity (such as MaAsLin2). However, one key message of our paper is that the DA tools can vary in performance: sometimes identifying no features as significant and other times identifying large proportions. Because of this we think it would be misleading to readers to suggest that there was a single tool that would be best to use. We realize that this

will be unsatisfying to not have a clear answer for many readers, but we believe it is a fair interpretation of our data.

Similarly, although we were partially able to explain what factors are associated with variation in tool performance, we do not have data on any specific applications that each tool would be best suited for (besides the general inference of which tools have higher specificity/sensitivity, which we highlight in the discussion). However, our analysis on the likelihood of identifying false positives did point out that some methods have inappropriate type I error rates when used on microbiome data. Furthermore, in our discussion we highlighted that microbiome researchers need to take further consideration into the normalization approaches they are using on their data and how those approaches impact the biological observations that they are making. For example, centred log-ratio approaches compare the abundances of microbes to the geometric mean abundance within the sample which in some cases can have a vastly different interpretation than when comparing proportions (e.g., 10.3389/fmicb.2017.02224). In our discussion we now highlight the importance of clearly describing how the results of certain tools should be interpreted differently than other tools that may have been used in similar papers.

Reviewers' Comments:

Reviewer #1:

Remarks to the Author:

Overall, the authors have addressed most of the concerns/comments adequately. It appears that the key aspects that this reviewer was hoping that the authors would address are clearly outside the scope of the authors' study since all that is being studied here is the differential abundance analysis regardless of the nature of the input data.

This is fine since each step of data analysis could benefit from scrutiny. For the scope of the study, the authors have provided a detailed and comprehensive report outlining the pros and cons of the various analysis tools available.

Reviewer #2:

Remarks to the Author:

The authors took the comments of all reviewers seriously, and improved their manuscript. I have several further concerns, listed below.

I203. "FDR-corrected p-values were used for methods that output p-values". I suppose this is the Benjamini-Hochberg correction, but then this should be mentioned and the publication cited. Also, I prefer the term "adjusted" p-values, but perhaps that's just me.

I499. "as input into receiver operator curves". What prediction method is used here? Logistic regression?

I608. "what percentage of tested ASVs was called as significant by each tool even when there is no difference expected between the sample groups." As in my previous comment: this is not the definition of the false discovery rate; and this is not the quantity being controlled by the Benjamini-Hochberg adjustment. The authors should give this quantity a new name.

I.760 Hawinkel et al also found excessive FDR for metagenomeSeq

Reviewer #3:

Remarks to the Author:

The authors responded adequately to my previous comments. I have no further comments.

Review Response Letter

Reviewer #1 (Remarks to the Author):

Overall, the authors have addressed most of the concerns/comments adequately. It appears that the key aspects that this reviewer was hoping that the authors would address are clearly outside the scope of the authors' study since all that is being studied here is the differential abundance analysis regardless of the nature of the input data.

This is fine since each step of data analysis could benefit from scrutiny. For the scope of the study, the authors have provided a detailed and comprehensive report outlining the pros and cons of the various analysis tools available.

Reviewer #2 (Remarks to the Author):

The authors took the comments of all reviewers seriously, and improved their manuscript. I have several further concerns, listed below.

l203. "FDR-corrected p-values were used for methods that output p-values". I suppose this is the Benjamini-Hochberg correction, but then this should be mentioned and the publication cited. Also, I prefer the term "adjusted" p-values, but perhaps that's just me.

We have addressed this comment by adding the type of p-value adjustment and citing the relevant paper.

l499. "as input into receiver operator curves". What prediction method is used here? Logistic regression?

We have updated our text to be clearer on how receiver operator curves were generated.

"For each ASV in a dataset the abundance of that ASV along with metadata groupings was used as input into the prediction function in the ROCR R package⁸⁰. Multiple different optimal abundance cut-offs were then used to classify samples based on the input ASVs abundance. Classifications were then compared to the true sample groupings to generate ROCs for each ASV within the 38 tested datasets."

l608. "what percentage of tested ASVs was called as significant by each tool even when there is no difference expected between the sample groups." As in my previous comment: this is not the definition of the false discovery rate; and this is not the quantity being controlled by the Benjamini-Hochberg adjustment. The authors should give this quantity a new name.

We have updated this text to indicate we are examining the false positive characteristics of the tools rather than their false discovery rates.

"Through this approach we were able to infer the false positive characteristics of each tool. In other words, we determined what percentage of tested ASVs was called as significant by each tool even when there is no difference expected between the sample groups."

l.760 Hawinkel et al also found excessive FDR for metagenomeSeq

We have added this reference to this line.

Reviewer #3 (Remarks to the Author):

The authors responded adequately to my previous comments. I have no further comments.